# Open Graph Benchmark:
# Datasets for Machine Learning on Graphs

**Weihua Hu**[1], **Matthias Fey**[2], **Marinka Zitnik**[3], **Yuxiao Dong**[4],
**Hongyu Ren**[1], **Bowen Liu**[5], **Michele Catasta**[1], **Jure Leskovec**[1]
[1]Department of Computer Science, [5]Chemistry, Stanford University
[2]Department of Computer Science, TU Dortmund University
[3]Department of Biomedical Informatics, Harvard University
[4]Microsoft Research, Redmond
`ogb@cs.stanford.edu`

**Steering Committee**
Regina Barzilay, Peter Battaglia, Yoshua Bengio, Michael Bronstein,
Stephan Günnemann, Will Hamilton, Tommi Jaakkola, Stefanie Jegelka,
Maximilian Nickel, Chris Re, Le Song, Jian Tang, Max Welling, Rich Zemel

## Abstract

We present the OPEN GRAPH BENCHMARK (OGB), a diverse set of challenging and realistic benchmark datasets to facilitate scalable, robust, and reproducible graph machine learning (ML) research. OGB datasets are large-scale, encompass multiple important graph ML tasks, and cover a diverse range of domains, ranging from social and information networks to biological networks, molecular graphs, source code ASTs, and knowledge graphs. For each dataset, we provide a unified evaluation protocol using meaningful application-specific data splits and evaluation metrics. In addition to building the datasets, we also perform extensive benchmark experiments for each dataset. Our experiments suggest that OGB datasets present significant challenges of scalability to large-scale graphs and out-of-distribution generalization under realistic data splits, indicating fruitful opportunities for future research. Finally, OGB provides an automated end-to-end graph ML pipeline that simplifies and standardizes the process of graph data loading, experimental setup, and model evaluation. OGB will be regularly updated and welcomes inputs from the community. OGB datasets as well as data loaders, evaluation scripts, baseline code, and leaderboards are publicly available at `https://ogb.stanford.edu`.

## 1   Introduction

Graphs are widely used for abstracting complex systems of interacting objects, such as social networks [30], knowledge graphs [63], molecular graphs [92], and biological networks [9], as well as for modeling 3D objects [75], manifolds [15], and source code [4]. Machine learning, especially deep learning, on graphs is an emerging field [15, 38]. Recently, significant methodological advances have been made in graph ML [35, 49, 84, 94, 100], which have produced promising results in applications from diverse domains [77, 99, 107].

How can we further advance research in graph ML? Historically, high-quality and large-scale datasets have played significant roles in advancing research, as exemplified by IMAGENET [23] and MS COCO [58] in computer vision, GLUE BENCHMARK [86] and SQUAD [69] in natural language processing, and LIBRISPEECH [64] and CHIME [10] in speech processing. However, in graph

ML research, commonly-used datasets and evaluation procedures present issues that may negatively impact future progress.

**Issues with current benchmarks**. Most of the frequently-used graph datasets are extremely small compared to graphs found in real applications (with more than 1 million nodes or 100 thousand graphs) [12, 43, 85, 87, 92, 99]. For example, the widely-used node classification datasets, CORA, CITESEER, and PUBMED [98], only have 2,700 to 20,000 nodes, the popular graph classification datasets from the TU collection [47, 95] only contain 200 to 5,000 graphs, and the commonly-used knowledge graph completion datasets, FB15K and WN18 [14], only have 15,000 to 40,000 entities. As models are extensively developed on these small datasets, the majority of them turn out to be not scalable to larger graphs [14, 49, 81, 83]. The small datasets also make it hard to rigorously evaluate data-hungry models, such as Graph Neural Networks (GNNs) [28, 34, 56, 94]. In fact, the performance of GNNs on these datasets is often unstable and nearly statistically identical to each other, due to the small number of samples the models are trained and evaluated on [29, 40].

Furthermore, there is no unified and commonly-followed experimental protocol. Different studies adopt their own dataset splits, evaluation metrics, and cross-validation protocols, making it challenging to compare performance reported across various studies [29, 31, 74]. In addition, many studies follow the convention of using random splits to generate training/test sets [14, 49, 94], which is not realistic or useful for real-world applications and generally leads to overly optimistic performance results [59]. An extensive discussion on the shortcomings of the current benchmarks is further provided in Appendix A.

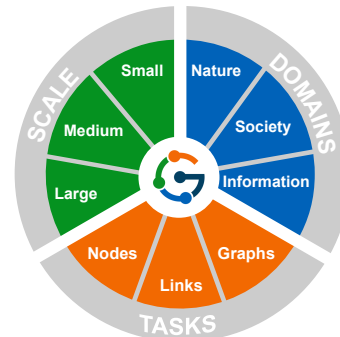

Figure 1: **OGB provides datasets that are diverse in scale, domains, and task categories.**

As a result, there is an urgent need for a comprehensive suite of real-world benchmarks that combine a diverse set of datasets of various sizes coming from different domains. Fixed data splits as well as evaluation metrics are important so that progress can be measured in a consistent and reproducible way. Last but not least, benchmarks need to provide different types of tasks, such as node classification, link prediction, and graph classification.

**Present work: OGB**. Here, we present the OPEN GRAPH BENCHMARK (OGB) with the goal of facilitating scalable, robust, and reproducible graph ML research. The premise of OGB is to develop a diverse set of challenging and realistic benchmark datasets that can empower the rigorous advancements in graph ML. As illustrated in Figure 1, the OGB datasets are designed to have the following three characteristics:

1. *Large scale.* The OGB datasets are orders-of-magnitude larger than existing benchmarks and can be categorized into three different scales (small, medium, and large). Even the "small" OGB graphs have more than 100 thousand nodes or more than 1 million edges, but are small enough to fit into the memory of a single GPU, making them suitable for testing computationally intensive algorithms. Additionally, OGB introduces "medium" (more than 1 million nodes or more than 10 million edges) and "large" (on the order of 100 million nodes or 1 billion edges) datasets, which can facilitate the development of scalable models based on mini-batching and distributed training.

2. *Diverse domains.* The OGB datasets aim to include graphs that are representative of a wide variety of domains, ranging from social and information networks to biological networks, molecular graphs, source code ASTs, and knowledge graphs. The broad coverage of domains in OGB empowers the development and demonstration of general-purpose models, and can be used to distinguish them from domain-specific techniques. Furthermore, for each dataset, OGB adopts domain-specific data splits (*e.g.*, based on time, species, molecular structure, GITHUB project, etc.) that are more realistic and meaningful than conventional random splits.

3. *Multiple task categories.* Besides data diversity, OGB supports three categories of fundamental graph ML tasks, *i.e.*, node, link, and graph property predictions, each of which requires the models to make predictions at different levels of graphs, *i.e.*, at the level of a node, link, and entire graph, respectively.

The currently-available OGB datasets are summarized in Table 1, and their graph statistics are provided in Table 2. Currently, OGB includes 15 diverse graph datasets, with at least 4 datasets

Table 1: **Summary of currently-available OGB datasets.** An OGB dataset, *e.g.*, `ogbg-molhiv`, is identified by its prefix (`ogbg-`) and its name (`molhiv`). The prefix specifies the category of the graph ML task, *i.e.*, node (`ogbn-`), link (`ogbl-`), or graph (`ogbg-`) property prediction. Datasets come from diverse domains: **Nature** domain includes biological networks and molecular graphs, **Society** domain includes academic graphs and e-commerce networks, and **Information** domain includes knowledge graphs. A realistic data split scheme is provided for each dataset, whose detail can be found in Appendices B, C, and D, for each dataset.

| Category | Name | Domain | Node Feat. | Edge Feat. | Directed | Hetero | #Tasks | Split Scheme | Split Ratio | Task Type | Metric |
|---|---|---|---|---|---|---|---|---|---|---|---|
| **Node** `ogbn-` | products | Society | ✔ | – | – | – | 1 | Sales rank | 8/2/90 | Multi-cls class. | Accuracy |
| | proteins | Nature | – | ✔ | – | – | 112 | Species | 65/16/19 | Binary class. | ROC-AUC |
| | arxiv | Society | ✔ | – | ✔ | – | 1 | Time | 54/18/28 | Multi-cls class. | Accuracy |
| | papers100M | Society | ✔ | – | ✔ | – | 1 | Time | 78/8/14 | Multi-cls class. | Accuracy |
| | mag | Infomation | ✔ | ✔ | ✔ | ✔ | 1 | Time | 85/9/6 | Multi-cls class. | Accuracy |
| **Link** `ogbl-` | ppa | Nature | ✔ | – | – | – | 1 | Throughput | 70/20/10 | Link prediction | Hits@100 |
| | collab | Society | ✔ | – | – | – | 1 | Time | 92/4/4 | Link prediction | Hits@50 |
| | ddi | Nature | – | – | – | – | 1 | Protein target | 80/10/10 | Link prediction | Hits@20 |
| | citation | Society | ✔ | – | ✔ | – | 1 | Time | 99/1/1 | Link prediction | MRR |
| | wikikg | Information | – | ✔ | ✔ | – | 1 | Time | 94/3/3 | KG completion | MRR |
| | biokg | Information | – | ✔ | ✔ | ✔ | 1 | Random | 94/3/3 | KG completion | MRR |
| **Graph** `ogbg-` | molhiv | Nature | ✔ | ✔ | – | – | 1 | Scaffold | 80/10/10 | Binary class. | ROC-AUC |
| | molpcba | Nature | ✔ | ✔ | – | – | 128 | Scaffold | 80/10/10 | Binary class. | AP |
| | ppa | Nature | – | ✔ | – | – | 1 | Species | 49/29/22 | Multi-class class. | Accuracy |
| | code | Information | ✔ | ✔ | ✔ | – | 1 | Project | 90/5/5 | Sub-token prediction | F1 score |

Table 2: **Statistics of currently-available OGB datasets.** The first 3 statistics are calculated over raw training/validation/test graphs. The last 4 graph statistics are calculated over the 'standardized' training graphs, where the graphs are first converted into undirected and unlabeled homogeneous graphs with duplicated edges removed. The SNAP library [53] is then used to compute the graph statistics, where the graph diameter is approximated by performing BFS from 1,000 randomly-sampled nodes. The MaxSCC ratio represents the fraction of nodes in the largest strongly connected component of the graph.

| Category | Name | Scale | #Graphs | Average #Nodes | Average #Edges | Average Node Deg. | Average Clust. Coeff. | MaxSCC Ratio | Graph Diameter |
|---|---|---|---|---|---|---|---|---|---|
| **Node** `ogbn-` | products | medium | 1 | 2,449,029 | 61,859,140 | 50.5 | 0.411 | 0.974 | 27 |
| | proteins | medium | 1 | 132,534 | 39,561,252 | 597.0 | 0.280 | 1.000 | 9 |
| | arxiv | small | 1 | 169,343 | 1,166,243 | 13.7 | 0.226 | 1.000 | 23 |
| | papers100M | large | 1 | 111,059,956 | 1,615,685,872 | 29.1 | 0.085 | 1.000 | 25 |
| | mag | medium | 1 | 1,939,743 | 25,582,108 | 21.7 | 0.098 | 1.000 | 6 |
| **Link** `ogbl-` | ppa | medium | 1 | 576,289 | 30,326,273 | 73.7 | 0.223 | 0.999 | 14 |
| | collab | small | 1 | 235,868 | 1,285,465 | 8.2 | 0.729 | 0.987 | 22 |
| | ddi | small | 1 | 4,267 | 1,334,889 | 500.5 | 0.514 | 1.000 | 5 |
| | citation | medium | 1 | 2,927,963 | 30,561,187 | 20.7 | 0.178 | 0.996 | 21 |
| | wikikg | medium | 1 | 2,500,604 | 17,137,181 | 12.2 | 0.168 | 1.000 | 26 |
| | biokg | small | 1 | 93,773 | 5,088,434 | 47.5 | 0.409 | 0.999 | 8 |
| **Graph** `ogbg-` | molhiv | small | 41,127 | 25.5 | 27.5 | 2.2 | 0.002 | 0.993 | 12.0 |
| | molpcba | medium | 437,929 | 26.0 | 28.1 | 2.2 | 0.002 | 0.999 | 13.6 |
| | ppa | medium | 158,100 | 243.4 | 2,266.1 | 18.3 | 0.513 | 1.000 | 4.8 |
| | code | medium | 452,741 | 125.2 | 124.2 | 2.0 | 0.0 | 1.000 | 13.5 |

for each task category. All the datasets are constructed by ourselves, except for `ogbn-products`, `ogbg-molpcba`, and `ogbg-molhiv`, whose graphs and target labels are adopted from Chiang *et al.* [17] and Wu *et al.* [92]. For these datasets, we resolve critical issues of the existing data splits by presenting more meaningful and standardized splits. OGB is a community-driven, open-source initiative. Over time, we plan to release new datasets and tasks, based on the input from the community.

In addition to building the graph datasets, we also perform extensive benchmark experiments for each dataset. Through the experiments and ablation studies, we highlight research challenges and opportunities provided by each dataset, especially on (1) scaling models to large graphs, and (2) improving *out-of-distribution* generalization performance under the realistic data split scenarios.



| OGB Graph Datasets | OGB Data Loader | Your ML Model | OGB Evaluator | OGB Leaderboards |
|:---:|:---:|:---:|:---:|:---:|
| (a) | (b) | (c) | (d) | (e) |

Figure 2: **Overview of the OGB pipeline: (a)** OGB provides realistic graph benchmark datasets that cover different prediction tasks (node, link, graph), are from diverse application domains, and are at different scales. **(b)** OGB fully automates dataset processing and splitting. That is, the OGB data loaders automatically download and process graphs, provide graph objects (compatible with PYTORCH [65] and its associated graph libraries, PYTORCH GEOMETRIC [33] and DEEP GRAPH LIBRARY [88]), and further split the datasets in a standardized manner. **(c)** After an ML model is developed, **(d)** OGB evaluates the model in a dataset-dependent manner, and outputs the model performance appropriate for the task at hand. Finally, **(e)** OGB provides public leaderboards to keep track of recent advances. Steps **(b)** and **(d)** are supported by our OGB Python package, whose usage is explained in Appendix E.

Finally, as illustrated in Figure 2, OGB presents an automated end-to-end graph ML pipeline that simplifies and standardizes the process of graph data loading, experimental setup, and model evaluation, in the same spirit as OpenML [32, 82]. Specifically, given the OGB datasets (a), the end-users can focus on developing their graph ML models (c) by using the OGB data loaders (b) and evaluators (d), both of which are provided by our OGB Python package (`https://github.com/snap-stanford/ogb`). OGB also hosts a public leaderboard (e) for publicizing state-of-the-art, reproducible graph ML research (`https://ogb.stanford.edu/docs/leader_overview`).

## 2 OGB Datasets and Benchmark Analyses: Overview

The goal of OGB is to catalyze graph ML research by providing realistic, diverse, and large-scale graph datasets with unified evaluation protocols. Table 1 summarizes the OGB datasets along with their graph types, prediction tasks, as well as evaluation protocols (data splits and evaluation metrics).

In the subsequent sections (Sections 3, 4, and 5), we detail currently-available datasets for each task category. Along with this, we provide an extensive benchmark analysis for each dataset, using representative node embedding models, GNNs, as well as recently-introduced mini-batch-based GNNs. We discuss our initial findings, and highlight research challenges and opportunities in: (1) scaling models to large graphs, and (2) improving *out-of-distribution* generalization under the realistic data splits. We repeat each experiment 10 times using different random seeds and report the mean and unbiased standard deviation of all training and test results corresponding to the best validation results. All code to reproduce our baseline experiments is publicly available at `https://github.com/snap-stanford/ogb/tree/master/examples` and is meant as a starting point to accelerate further research on our proposed datasets. We refer the interested reader to our code base for the details of model architectures and hyper-parameter settings.

Finally, we highlight the diversity of our graph datasets by comparing their basic graph statistics in Table 2. Importantly, we observe the diversity in *graph structure*, beyond the diversity in the dataset scales. For example, comparing the average node degrees, we see that biology-related graphs (*e.g.*, `ogbn-proteins`, `ogbl-ddi`, `ogbl-ppa`, `ogbg-ppa`) are much denser than the social and information networks. The other statistics (average clustering coefficient and graph diameter) also vary significantly across different datasets. These differences in graph structure result in the inherent difference in how information propagates in the graphs, which can significantly affect the behavior of many graph ML models such as GNNs and random-walk-based node embeddings [93]. For the graph property prediction datasets, it is worth highlighting the diversity of graph sizes (the number of nodes and edges *per graph*), ranging from small molecular graphs (`ogbg-molhiv` and `ogbg-molpcba`), to medium-sized source code ASTs (`ogbg-code`), up to large and dense protein-protein association subgraphs (`ogbg-ppa`). Overall, the diversity in graph characteristics originates from the diverse application domains and is crucial to evaluate the versatility of graph ML models.

# 3 OGB Node Property Prediction

We currently provide 5 datasets, adopted from diverse application domains, for predicting the properties of individual nodes. Specifically, `ogbn-products` is an Amazon products co-purchasing network [12] originally developed by Chiang *et al.* [17]. The `ogbn-arxiv`, `ogbn-mag`, and `ogbn-papers100M` datasets are extracted from the Microsoft Academic Graph (MAG) [87], with different scales, tasks, and include both homogeneous and heterogeneous graphs. Specifically, `ogbn-arxiv` is a paper citation network of ARXIV papers, `ogbn-mag` is a heterogeneous academic graph containing different node types (papers, authors, institutions, and topics) and their relations, and `ogbn-papers100M` is an extremely large paper citation network from the entire MAG with more than *100 million nodes* and *1 billion edges*. The `ogbn-proteins` dataset is a protein-protein association network [80]. Below we present the `ogbn-products` dataset and its baseline experiments. Due to space constraints, we present all the datasets comprehensively in Appendix B.

**The `ogbn-products` dataset**. This dataset is an undirected and unweighted graph, representing an Amazon product co-purchasing network [12]. Nodes represent products sold in Amazon, and edges between two products indicate that the products are purchased together. The graphs, target labels, and node features are generated following Chiang *et al.* [17], where node features are dimensionality-reduced bag-of-words of the product descriptions. Our contribution, when adopting the dataset in OGB, is to resolve its critical data split issue by presenting a more realistic and challenging split (see below).

**Prediction task**. The task is to predict the category of a product in a multi-class classification setup, where the 47 top-level categories are used for target labels.

**Dataset splitting**. We consider a more challenging and realistic dataset splitting that differs from the one used in Chiang *et al.* [17]. Instead of *randomly* assigning 90% of the nodes for training and 10% of the nodes for testing (without a validation set), we use the *sales ranking* (popularity) to split nodes into training/validation/test sets. Specifically, we sort the products according to their sales ranking and use the top 8% for training, next top 2% for validation, and the rest for testing. This is a more challenging splitting procedure that closely matches the real-world application where manual labeling is prioritized to important nodes in the network and ML models are subsequently used to make predictions on less important ones.

**Baselines**. We perform an extensive empirical study, including the representative node embedding model, GNNs, and as well as recently-introduced mini-batch-based GNN models, as baselines.

- **MLP**: A multilayer perceptron (MLP) predictor that uses the given raw node features directly as input. Graph structure information is not utilized.
- **NODE2VEC**: An MLP predictor that uses as input the concatenation of the raw node features and NODE2VEC embeddings [35, 66].
- Full-batch GNNs: **GCN** [49] and **GRAPHSAGE** (mean pool) [37].
- Mini-batch training of GNNs based on **NEIGHBORSAMPLING** [37], **CLUSTERGCN** [17] and **GRAPHSAINT** [103].

Note that the full-batch GCN and GRAPHSAGE are GPU memory-intensive for large graphs as all the node embeddings need to loaded onto GPU all at once. The mini-batch training techniques, NEIGHBORSAMPLING, CLUSTERGCN, and GRAPHSAINT, do not suffer from this issue and are more GPU memory-efficient. All models are trained with a fixed hidden dimensionality of 256, a fixed number of three layers, and a tuned dropout ratio $\in \{0.0, 0.5\}$.

**Results and discussion**. Our benchmarking results in Table 3 show that the highest test performances are attained by GNNs, while the MLP baseline that solely relies on a product's description is not sufficient for accurately predicting the category of a product. Even with the GNNs, we observe the huge generalization gap[1], which can be explained by differing node distributions across the splits, as visualized in Figure 3. This is in stark contrast with the conventional *random split* used by Chiang *et al.* [17]. Even with the same split ratio (8/2/88), we find GRAPHSAGE already achieves $88.20_{\pm 0.08}\%$ test accuracy with only $\approx 1$ percentage points of generalization gap. These results indicate that the realistic split is much more challenging than the random split and offer an important opportunity to improve *out-of-distribution* generalization.

Table 3: **Results for `ogbn-products`.**
[†]Requires a GPU with 33GB of memory.

| Method | Accuracy (%) | | |
|---|---|---|---|
| | Training | Validation | **Test** |
| MLP | $84.03_{\pm0.93}$ | $75.54_{\pm0.14}$ | $61.06_{\pm0.08}$ |
| NODE2VEC | $93.39_{\pm0.10}$ | $90.32_{\pm0.06}$ | $72.49_{\pm0.10}$ |
| GCN[†] | $93.56_{\pm0.09}$ | $92.00_{\pm0.03}$ | $75.64_{\pm0.21}$ |
| GRAPHSAGE[†] | $94.09_{\pm0.05}$ | $\mathbf{92.24}_{\pm0.07}$ | $78.50_{\pm0.14}$ |
| NEIGHBORSAMPLING | $92.96_{\pm0.07}$ | $91.70_{\pm0.09}$ | $78.70_{\pm0.36}$ |
| CLUSTERGCN | $93.75_{\pm0.13}$ | $92.12_{\pm0.09}$ | $\mathbf{78.97}_{\pm0.33}$ |
| GRAPHSAINT | $92.71_{\pm0.14}$ | $91.62_{\pm0.08}$ | $\mathbf{79.08}_{\pm0.24}$ |

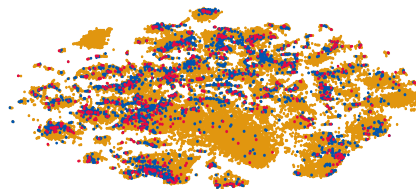

Figure 3: T-SNE visualization of training/validation/test nodes in `ogbn-products`.

Table 3 also shows that the recent mini-batch-based GNNs[2] give promising results, even slightly outperforming the full-batch version of GRAPHSAGE that does not fit into ordinary GPU memory. The improved performance can be attributed to the regularization effects of mini-batch noise and edge dropout [71]. Nevertheless, the mini-batch GNNs have been much less explored compared to the full-batch GNNs due to the prevalent use of the extremely small benchmark datasets such as CORA and CITESEER. As a result, many important questions remain open, *e.g.*, what mini-batch training methods can induce the best regularization effect, and how to allow mini-batch training for advanced GNNs that rely on large receptive-field sizes [50, 54, 93], since the current mini-batch methods are rather limited by the number of nodes from which they aggregate information. Overall, `ogbn-products` is an ideal benchmark dataset for the field to move beyond the extremely small graph datasets and to catalyze the development of scalable mini-batch-based graph models with improved *out-of-distribution* prediction accuracy.

In Appendix B.4, we present `ogbn-papers100M`, which is even larger and is meant to push the scalability to gigantic web-scale graphs in the real world.

# 4 OGB Link Property Prediction

We currently provide 6 datasets, adopted from diverse application domains, for predicting the properties of links (pairs of nodes). Specifically, `ogbl-ppa` is a protein-protein association network [80], `ogbl-collab` is an author collaboration network [87], `ogbl-ddi` is a drug-drug interaction network [90], `ogbl-citation` is a paper citation network [87], `ogbl-biokg` is a heterogeneous knowledge graph compiled from a large number of biomedical repositories, and `ogbl-wikikg` is a Wikidata knowledge graph [85]. Below we present the `ogbl-wikikg` dataset and its baseline experiments. Due to space constraints, we present all the datasets comprehensively in Appendix C.

**The `ogbl-wikikg` dataset**. This dataset is a Knowledge Graph (KG) extracted from the Wikidata knowledge base [85]. It contains a set of triplet edges (head, relation, tail), capturing the different types of relations between entities in the world, *e.g.*, $Canada \xrightarrow{citizen} Hinton$. We retrieve all the relational statements in Wikidata and filter out rare entities. Our KG contains 2,500,604 entities and 535 relation types.

**Prediction task**. The task is to predict new triplet edges given the training edges. The evaluation metric follows the standard filtered metric widely used in KGs [14, 78, 81, 96]. Specifically, we corrupt each test triplet edge by replacing its head or tail with randomly-sampled 1,000 negative entities (500 for head and 500 for tail), while ensuring the resulting triplets do not appear in the KG. The goal is to rank the true head (or tail) entities higher than the negative entities, which is measured by the Mean Reciprocal Rank (MRR).

**Dataset splitting**. We split the triplets according to time, simulating a realistic KG completion scenario that aims to fill in missing triplets that are not present at a certain timestamp. Specifically, we downloaded Wikidata at three different time stamps[3] (May, August, and November of 2015), and construct three KGs, where we only retain entities and relation types that appear in the earliest May

Table 4: **Results for `ogbl-wikikg`.**
†Requires a GPU with 48GB of memory.

| Method | MRR | | |
|---|---|---|---|
| | Training (Unfiltered) | Validation (Filtered) | **Test** (Filtered) |
| TRANSE | $0.3326_{\pm 0.0041}$ | $0.2314_{\pm 0.0035}$ | $0.2535_{\pm 0.0036}$ |
| DISTMULT | $0.4131_{\pm 0.0057}$ | $0.3142_{\pm 0.0066}$ | $0.3434_{\pm 0.0079}$ |
| COMPLEX | $0.4605_{\pm 0.0020}$ | $0.3612_{\pm 0.0063}$ | $0.3877_{\pm 0.0051}$ |
| ROTATE | $0.3469_{\pm 0.0055}$ | $0.2366_{\pm 0.0043}$ | $0.2681_{\pm 0.0047}$ |
| TRANSE ($6\times$dim)† | $0.6491_{\pm 0.0022}$ | $\mathbf{0.4587_{\pm 0.0031}}$ | $\mathbf{0.4536_{\pm 0.0028}}$ |
| DISTMULT ($6\times$dim)† | $0.4339_{\pm 0.0011}$ | $0.3403_{\pm 0.0009}$ | $0.3612_{\pm 0.0030}$ |
| COMPLEX ($6\times$dim)† | $0.4712_{\pm 0.0045}$ | $0.3787_{\pm 0.0036}$ | $0.4028_{\pm 0.0033}$ |
| ROTATE ($6\times$dim)† | $0.6084_{\pm 0.0025}$ | $0.3613_{\pm 0.0031}$ | $0.3626_{\pm 0.0041}$ |

KG. We use the triplets in the May KG for training, and use the additional triplets in the August and November KGs for validation and test, respectively. Note that our dataset split is different from the existing Wikidata KG dataset that adopts a conventional random split [89], which does not reflect the practical usage of KG completion.

**Baselines**. We consider the four representative KG embedding models: **TRANSE** [14], **DISTMULT** [96], **COMPLEX** [81], and **ROTATE** [78]. For KGs with many entities and relations, the embedding dimensionality can be limited by the available GPU memory, as the embeddings need to be loaded into GPU all at once. We therefore choose the dimensionality such that training can be performed on a fixed-budget of GPU memory. Our training procedure follows Sun *et al.* [78], where we perform negative sampling and use margin-based logistic loss for the loss function.

**Results and discussion**. Our benchmark results are provided in Table 4, where the upper-half baselines are implemented on a single commodity GPU with 11GB memory, while the bottom-half baselines are implemented on a high-end GPU with 48GB memory.[4] Training MRR in Table 4 is an *unfiltered* metric,[5] as filtering is computationally expensive for the large number of training triplets.

First, we see from the upper-half of Table 4 that when the limited embedding dimensionality is used, COMPLEX performs the best among the four baselines. With the increased dimensionality, all four models are able to achieve higher MRR on training, validation and test sets, as seen from the bottom-half of Table 4. This suggests the importance of using a sufficient large embedding dimensionality to achieve good performance in this dataset. Interestingly, although TRANSE performs the worst with the limited dimensionality, it obtains the best performance with the increased dimensionality. Nevertheless, the extremely low test MRR[6] suggests that our realistic KG completion dataset is highly non-trivial. It presents a realistic generalization challenge of *discovering* new triplets based on existing ones, which necessitates the development of KG models with more robust and generalizable reasoning capability. Furthermore, this dataset presents an important challenge of effectively scaling embedding models to large KGs—naïvely training KG embedding models with reasonable dimensionality would require a high-end GPU, which is extremely costly and not scalable to even larger KGs. A promising approach to improve scalability is to distribute training across multiple commodity GPUs [52, 105, 106]. A different approach is to share parameters across entities and relations, so that a smaller number of embedding parameters need to be put onto the GPU memory at once.

Table 5: **Results for `ogbg-molhiv`.**

| Method | Add. Feat. | Virt. Node | ROC-AUC (%) Training | Validation | Test |
|---|---|---|---|---|---|
| GCN | ✗ | ✔ | $88.65_{\pm1.01}$ | $83.73_{\pm0.78}$ | $74.18_{\pm1.22}$ |
| | ✔ | ✗ | $88.65_{\pm2.19}$ | $82.04_{\pm1.41}$ | $76.06_{\pm0.97}$ |
| | ✔ | ✔ | $90.07_{\pm4.69}$ | $83.84_{\pm0.91}$ | $75.99_{\pm1.19}$ |
| GIN | ✗ | ✔ | $93.89_{\pm2.96}$ | $84.10_{\pm1.05}$ | $75.20_{\pm1.30}$ |
| | ✔ | ✗ | $88.64_{\pm2.54}$ | $82.32_{\pm0.90}$ | $75.58_{\pm1.40}$ |
| | ✔ | ✔ | $92.73_{\pm3.80}$ | $\mathbf{84.79_{\pm0.68}}$ | $\mathbf{77.07_{\pm1.49}}$ |

Table 6: **Results for `ogbg-molpcba`.**

| Method | Add. Feat. | Virt. Node | AP (%) Training | Validation | Test |
|---|---|---|---|---|---|
| GCN | ✗ | ✔ | $36.25_{\pm0.71}$ | $23.88_{\pm0.22}$ | $22.91_{\pm0.37}$ |
| | ✔ | ✗ | $28.04_{\pm0.58}$ | $20.59_{\pm0.33}$ | $20.20_{\pm0.24}$ |
| | ✔ | ✔ | $38.25_{\pm0.50}$ | $24.95_{\pm0.42}$ | $24.24_{\pm0.34}$ |
| GIN | ✗ | ✔ | $45.70_{\pm0.61}$ | $27.54_{\pm0.25}$ | $26.61_{\pm0.17}$ |
| | ✔ | ✗ | $37.05_{\pm0.31}$ | $23.05_{\pm0.27}$ | $22.66_{\pm0.28}$ |
| | ✔ | ✔ | $46.96_{\pm0.57}$ | $\mathbf{27.98_{\pm0.25}}$ | $\mathbf{27.03_{\pm0.23}}$ |

# 5 OGB Graph Property Prediction

We currently provide 4 datasets, adopted from 3 distinct application domains, for predicting the properties of entire graphs or subgraphs. Specifically, `ogbg-molhiv` and `ogbg-molpcba` are molecular graphs originally curated by Wu *et al.* [92], `ogbg-ppa` is a set of protein-protein association subgraphs [108], and `ogbg-code` is a collection of ASTs of source code [43]. Below we present the `ogbg-molhiv` and `ogbg-molpcba` datasets and their baseline experiments. Due to space constraints, we present all the datasets comprehensively in Appendix D.

**The `ogbg-molhiv` and `ogbg-molpcba` datasets**. These datasets are two molecular property prediction datasets adopted from the MOLECULENET [92], and are among the largest of the MOLECULENET datasets. Besides the two main molecule datasets, we also provide the 10 other MOLECULENET datasets, which are summarized and benchmarked in Appendix F. These datasets can be used to stress-test molecule-specific methods [46, 97] and transfer learning [40]. All the molecules are pre-processed using RDKIT [51]. Each graph represents a molecule, where nodes are atoms, and edges are chemical bonds. Input node features are 9-dimensional, containing atomic number and chirality, as well as other *additional* atom features such as formal charge and whether the atom is in the ring. Input edge features are 3-dimensional, containing bond type, bond stereochemistry as well as an *additional* bond feature indicating whether the bond is conjugated. Note that the above additional features are not needed to uniquely identify molecules, and are not adopted in the previous work [40, 44]. In the experiments, we perform an ablation study on the molecule features and find that including the additional features improves generalization performance.

**Prediction task**. The task is to predict the target molecular properties as accurately as possible, where the molecular properties are cast as binary labels, *e.g.*, whether a molecule inhibits HIV virus replication or not. For evaluation metric, we closely follow Wu *et al.* [92]. Specifically, for `ogbg-molhiv`, we use ROC-AUC for evaluation. For `ogbg-molpcba`, as the class balance is extremely skewed (only 1.4% of data is positive) and the dataset contains multiple classification tasks, we use the Average Precision (AP) averaged over the tasks as the evaluation metric.

**Dataset splitting**. We adopt the *scaffold splitting* procedure that splits the molecules based on their two-dimensional structural frameworks. The scaffold splitting attempts to separate structurally different molecules into different subsets, which provides a more realistic estimate of model performance in prospective experimental settings. The scaffold splitting was originally proposed by Wu *et al.* [92] and has been adopted by the follow-up works [40, 44, 70, 97]; however, the precise implementation differs significantly across works, making their results not directly comparable to each other. In OGB, we aim to standardize the scaffold split by adopting its most challenging version where test molecules are maximally diverse.

**Baselines**. We consider the two representative GNNs: **GCN** [49] and **GIN** [94]. We additionally consider augmenting the models with VIRTUAL NODES, where message-passing is performed over an augmented graph with an additional node that is connected to all nodes in the original graph [34, 44, 55]. We use 5-layer GNNs, average graph pooling, a hidden dimensionality of 300, and a tuned dropout ratio of {0, 0.5}. To include edge features, we follow Hu *et al.* [40] and add transformed edge features into the incoming node features.

**Results and discussion**. Benchmarking results are given in Tables 5 and 6. We see that GIN with both additional features and VIRTUAL NODES provides the best performance in the two datasets. In Appendix F, we show that even for the other MOLECULENET datasets, the additional features

consistently improve generalization performance. In OGB, we therefore include the additional node/edge features in our molecular graphs.

We further report the performance on the random splitting, keeping the split ratio the same as the scaffold splitting. We find the random split to be much easier than scaffold split. On random splits of `ogbg-molhiv` and `ogbg-molpcba`, the best GIN achieves the ROC-AUC of $82.73_{\pm 2.02}\%$ (5.66 percentage points higher than scaffold) and AP of $34.40_{\pm 0.90}\%$ (7.37 percentage points higher than scaffold), respectively. The same trend holds true for the other MOLECULENET datasets, *e.g.*, the best GIN performance on the random split of `ogbg-moltox21` is $86.03_{\pm 1.37}\%$, which is 8.46 percentage points higher than that of the best GIN for the scaffold split ($77.57_{\pm 0.62}\%$ ROC-AUC). These results highlight the challenges of the scaffold split compared to the random split, and opens up a fruitful research opportunity to increase the out-of-distribution generalization capability of GNNs.

# 6  Conclusions

To enable scalable, robust, and reproducible graph ML research, we introduce the Open Graph Benchmark (OGB)—a diverse set of realistic graph datasets in terms of scales, domains, and task categories. We employ realistic data splits for the given datasets, driven by application-specific use cases. Through extensive benchmark experiments, we highlight that the OGB datasets present significant challenges for ML models to handle large-scale graphs and make accurate prediction under the realistic data splitting scenarios. Altogether, OGB presents fruitful opportunities for future research to push the frontier of graph ML.

OGB is an open-source initiative that provides ready-to-use datasets as well as their data loaders, evaluation scripts, and public leaderboards. We hereby invite the community to develop and contribute state-of-the-art graph ML models at `https://ogb.stanford.edu`.

## Broader Impact

We expect the Open Graph Benchmark (OGB) to have a significant impact on fundamental graph ML research as well as many of its application domains. We also discuss a potential negative impact.

### Impact on Graph ML Research

Historically, high-quality and large-scale datasets have played significant roles in advancing research fields (*e.g.*, IMAGENET [23], MS-COCO [58], GLUE benchmark [86], SQUAD [69]). The amount of impact these datasets have brought is enormous, leading to the significant methodological advancements in the respective fields [24, 39].

We expect OGB to be a standard benchmark in graph ML, contributing to the significant advancements of the field. To this end, our datasets are carefully designed to address the two major drawbacks of current graph benchmark datasets, namely (1) small dataset sizes, and (2) unrealistic random splits. Overall, OGB provides a set of diverse, realistic, and large-scale graph datasets to facilitate the development of graph ML models that are scalable and generalizable under realistic data splits, both of which are crucial in practice.

In addition, OGB aims to address the fundamental problem of reproducibility in graph ML research. We promote the reproducibility by standardizing the research pipeline, as illustrated in Figure 2, and provide official leaderboards, for which public code is mandatory to make a submission. Altogether, OGB incentivises researchers to release their code, and allows researchers to easily compare different models under equal settings.

### Impact on Diverse Application Domains

In OGB, we have curated graph datasets that are relevant to a variety of practical and realistic application domains, including science (*e.g.*, biology, chemistry), knowledge graphs, academic graphs, and source code ASTs. For example, we provided a biomedical knowledge graph (`ogbl-biokg`), where algorithmic advances on this dataset can be immediately translated into solutions for problems in precision medicine. In another example, we provide a dataset of 450K molecular graphs (`ogbg-molpcba`) that have direct implications for drug discovery. In academic domains, we

prepared a variety of prediction tasks (*e.g.*, recommending missing citations as well as future collaborations, predicting paper categories and venues, etc.), solving which can lead to improved scholarly efficiency and to better organization of academic knowledge. In technological domains, OGB includes a dataset of source code snippets (`ogbg-code`). The development of graph ML models on this dataset can lead to exciting applications for advanced code analysis and retrieval.

To further increase the impact of OGB, all of our datasets are mapped to real entities in the world. For example, each node in the drug-drug interaction network (`ogbl-ddi`) is mapped to a unique Drug ID in DrugBank [90], each molecule in the molecule datasets (`ogbg-mol*`) is mapped to a SMILES string that uniquely identifies the molecule, and each node in the paper citation networks (`ogbn-arxiv` and `ogbn-papers100M`) is mapped into a real research paper indexed by the Microsoft Academic Graph [87]. Such mappings to real entities allow researchers to draw scientific insight and to augment the graphs with external information.

**Potential Negative Impact**

If OGB becomes the standard de-facto benchmark for graph ML, one potential negative impact is that OGB might contribute to narrowing down the scope of future papers to the tasks and dataset types that have been included in OGB. In order to avoid such a negative impact, we will regularly update our datasets and tasks, based on the input from the community.

# Acknowledgements

We thank Adrijan Bradaschia and Rok Sosic for their help in setting up the server and website. We also thank Emma Pierson and Shigeru Maya for their suggestions on the paper writing. Finally, we thank the entire community of graph ML for providing valuable feedback to improve OGB. Weihua Hu is supported by Funai Overseas Scholarship and Masason Foundation Fellowship. Matthias Fey is supported by the German Research Association (DFG) within the Collaborative Research Center SFB 876 "Providing Information by Resource-Constrained Analysis", project A6. Marinka Zitnik is in part supported by NSF IIS-2030459. We gratefully acknowledge the support of DARPA under Nos. FA865018C7880 (ASED), N660011924033 (MCS); ARO under Nos. W911NF-16-1-0342 (MURI), W911NF-16-1-0171 (DURIP); NSF under Nos. OAC-1835598 (CINES), OAC-1934578 (HDR), CCF-1918940 (Expeditions), IIS-2030477 (RAPID); Stanford Data Science Initiative, Wu Tsai Neurosciences Institute, Chan Zuckerberg Biohub, Amazon, Boeing, JPMoran Chase, Docomo, Hitachi, JD.com, KDDI, NVIDIA, Dell. Jure Leskovec is a Chan Zuckerberg Biohub investigator.

## Footnotes

[1]Defined by the difference between training and test accuracy.

[2]The GRAPHSAGE architecture is used for neighbor aggregation.

[3]Available at `https://archive.org/search.php?query=creator%3A%22Wikidata+editors%22`

[4]Given a fixed 11GB GPU memory budget, we adopt 100-dimension embeddings for DISTMULT and TRANSE. Since ROTATE and COMPLEX require the entity embeddings with the real and imaginary parts, we train these two models with the dimensionality of 50 for each part. On the other hand, on the high-end GPU with 48GB memory, we are able to train all the models with $6\times$ larger embedding dimensionality.

[5]This means that the training MRR is computed by ranking against randomly-selected negative entities without filtering out triplets that appear in KG. The unfiltered metric has the systematic bias of being smaller than the filtered counterpart (computed by ranking against "true" negative entities, *i.e.*, the resulting triplets do not appear in the KG) [14].

[6]Note that our test MRR on `ogbl-wikikg` is computed using only 500 negative entities per triplet, which is much less than the number of negative entities used to compute MRR in the existing KG datasets, such as FB15K and FB15K-237 (around 15,000 negative entities). Nevertheless, ROTATE gives either lower or comparable test MRR on `ogbl-wikikg` compared to FB15K and FB15K-237 [78].

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
