[Supplementary Material]

# Appendices

## A  Shortcomings of Current Benchmarks

We first review commonly-used graph benchmarks and discuss the current state of the field. We organize the discussion around three categories of graph ML tasks: predictions at the level of nodes, links, and graphs.

**Node property prediction**. Currently, the three graphs (CORA, CITESEER and PUBMED) proposed in Yang *et al.* [98] have been widely used as semi-supervised node classification datasets, particularly for evaluating GNNs. The sizes of these graphs are rather small, ranging from 2,700 to 20,000 nodes. Recent studies suggest that datasets at this small scale can be solved quite well with simple GNN architectures [74, 91], and the performance of different GNNs on these datasets is nearly statistically identical [29, 40]. Furthermore, there is no consensus on the splitting procedures for these datasets, which makes it hard to fairly compare different model designs [74]. Finally, a recent study [109] shows that these datasets have some fundamental data quality issues. For example, in CORA, 42% of the nodes leak information between their features and labels, and 1% of the nodes are duplicated. The situation for CITESEER is even worse, with leakage rates of 62% and duplication rates of 5%.

Some recent works in graph ML have proposed relatively large datasets, such as PPI (56,944 nodes), REDDIT (334,863 nodes) [38] and AMAZON2M (2,449,029 nodes) [17]. However, there exist some inherent issues with the proposed data splits. Specifically, 83%, 65% and 90% of the nodes are used for training in the PPI, REDDIT and AMAZON2M datasets, respectively, which results in an artificially small distribution shift across the training/validation/test sets. Consequently, as may be expected, the performance improvements on these benchmarks have quickly saturated. For example, recent GNN models [17, 103] can already yield F1 scores of 99.5 for PPI and 97.0 for REDDIT, and 90.4% accuracy for AMAZON2M, with extremely small generalization gaps between training and test accuracy. Finally, it is also practically required for graph ML models to handle web-scale graphs (beyond 100 million nodes and 1 billion edges) in industrial applications [99]. However, to date, there have been no publicly available graph datasets of such scale with label information.

In summary, several factors (*e.g.*, size, leakage, splits, etc.) associated with the current use of existing datasets make them unsuitable as benchmark datasets for graph ML.

**Link property prediction**. Broadly, there are two lines of efforts for the link-level task: link prediction in homogeneous networks [57, 104] and relation completion in (heterogeneous) knowledge graphs [14, 41, 63]. There are several problems with the current benchmark datasets in these areas.

First, representative datasets are either extremely small or do not come with input node features. For link prediction, while the well-known recommender system datasets used in Berg *et al.* [11] include node features, their sizes are very small, with the largest having only 6,000 nodes. On the other hand, although the Open Academic Graph (OAG) used in Qiu *et al.* [68] comprises tens of millions of nodes, there are no associated node features. Regarding the knowledge graph completion, the widely-used dataset, FB15K, is very small, containing only 14,951 entities, which is a tiny subset of the original Freebase knowledge graph with more than 50 million entities [13].

Second, similar to the node-level task, random splits are predominantly used in link-level prediction [14, 35]. The random splits are not realistic in many practical applications such as friend recommendation in social networks, in which test edges (friend relations *after* a certain timestamp) naturally follow a different distribution from training edges (friend relations *before* a certain timestamp).

Finally, the existing datasets are mostly oriented to applications in recommender systems, social media and knowledge graphs, in which the graphs are typically very sparse. This may result in techniques specialised for sparse link inference that are not generalizable to domains with dense graphs, such as the protein-protein association graphs and drug-drug interaction networks typically found in biology and medicine [20, 67, 79, 80, 90]. Very recently, Sinha *et al.* [76] proposed a synthetic link prediction benchmark to diagnose model's logical generalization capability. Their focus is on synthetic tasks, which is complementary to OGB that focuses on realistic tasks.

**Graph property prediction**. Graph-level prediction tasks are found in important applications in natural sciences, such as predicting molecular properties in chemistry [28, 34, 40], where molecules are naturally represented as molecular graphs.

In graph classification, the most widely-used graph-level datasets from the TU collection [47] are known to have many issues, such as small sizes (*i.e.*, most of the datasets only contain less than 1,000 graphs),[7] unrealistic settings (*e.g.*, no bond features for molecules), random data splits, inconsistent evaluation protocols, and isomorphism bias [45]. A very recent attempt [29] to address these issues mainly focuses on benchmarking ML models, specifically the building blocks of GNNs, rather than developing application-oriented realistic datasets. In fact, five out of the six proposed datasets are synthetic.

Recent work in graph ML [40, 44] has started to adopt MOLECULENET [92] which contains a set of realistic and large-scale molecular property prediction datasets. However, there is limited consensus in the dataset splitting and molecular graph features, making it hard to compare different models in a fair manner. OGB adopts the MOLECULENET datasets, while providing unified dataset splits as well as the molecular graph features that are found to provide favorable performance over naïve features.

Beyond molecular graphs, OGB also provides graphs from other domains, such as biological networks and Abstract Syntax Tree (AST) representations of source code. These types of graphs exhibit distinct characteristics from molecular graphs, enabling the evaluation of the versatility of graph ML models.

## B  Comprehensive Descriptions of OGB Node Property Prediction

We currently provide 5 datasets, adopted from 3 different application domains, for predicting the properties of individual nodes. Specifically, `ogbn-products` is an Amazon products co-purchasing network [12] originally developed by Chiang *et al.* [17] (*cf.* Appendix B.1). The `ogbn-arxiv`, `ogbn-mag`, and `ogbn-papers100M` datasets are extracted from the Microsoft Academic Graph (MAG) [87], with different scales, tasks, and include both homogeneous and heterogeneous graphs. Specifically, `ogbn-arxiv` is a paper citation network of arXiv papers (*cf.* Appendix B.3), `ogbn-mag` is a heterogeneous academic graph containing different node types (papers, authors, institutions, and topics) and their relations (*cf.* Appendix B.5), and `ogbn-papers100M` is an extremely large paper citation network from the entire MAG with more than 100 million nodes and 1 billion edges (*cf.* Appendix B.4). The `ogbn-proteins` dataset is a protein-protein association network [80] (*cf.* Appendix B.2).

The 5 datasets exhibit highly diverse graph statistics, as shown in Table 2. Notably, the biological network, `ogbn-proteins`, is much denser than the social/information networks as can be observed from its large average node degree and small graph diameter. On the other hand, the co-purchasing network, `ogbn-products`, has more clustered graph structure than the other datasets, as can be seen from its large average clustering coefficient. Finally, the heterogeneous academic graph, `ogbn-mag`, exhibits rather interesting graph characteristics, simultaneously having small average node degree, clustering coefficient, and graph diameter.

**Baselines**. We consider the following representative models as our baselines unless otherwise specified.

- **MLP**: A multilayer perceptron (MLP) predictor that uses the given raw node features directly as input. Graph structure information is not utilized.
- **NODE2VEC**: An MLP predictor that uses as input the concatenation of the raw node features and NODE2VEC embeddings [35, 66].
- **GCN**: Full-batch Graph Convolutional Network [49].
- **GRAPHSAGE**: Full-batch GraphSAGE [37], where we adopt the mean pooling variant and a simple skip connection to preserve central node features.
- **NEIGHBORSAMPLING** (optional): A mini-batch training technique of GNNs [37] that samples neighborhood nodes when performing aggregation.
- **CLUSTERGCN** (optional): A mini-batch training technique of GNNs [17] that partitions the graphs into a fixed number of subgraphs and draws mini-batches from them.
- **GRAPHSAINT** (optional): A mini-batch training technique of GNNs [103] that samples subgraphs via a random walk sampler.

The three mini-batch GNN training, NEIGHBORSAMPLING, CLUSTERGCN, and GRAPHSAINT, are explored only for graph datasets where full-batch GCN/GRAPHSAGE did not fit into the common GPU memory size of 11GB. The mini-batch GNNs are more GPU memory-efficient than the full-batch GNNs because they first partition and sample the graph into subgraphs. Hence, in order to train the network, they require only a small amount of nodes to be loaded into the GPU memory at each mini-batch. Inference is then performed on the whole graph without GPU usage. To choose the architecture for the mini-batch GNNs, we first run full-batch GCN and GRAPHSAGE on an NVIDIA Quadro RTX 8000 with 48GB of memory, and then adopt the best performing full-batch GNN architecture for the mini-batch GNNs. All models are trained with a fixed hidden dimensionality of 256, a fixed number of two or three layers, and a tuned dropout ratio $\in \{0.0, 0.5\}$.

### B.1  `ogbn-products`: Amazon Products Co-purchasing Network

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

## B.2 `ogbn-proteins`: Protein-Protein Association Network

The `ogbn-proteins` dataset is an undirected, weighted, and typed (according to species) graph. Nodes represent proteins, and edges indicate different types of biologically meaningful associations between proteins, *e.g.*, physical interactions, co-expression or homology [18, 80]. All edges come with 8-dimensional features, where each dimension represents the strength of a single association type and takes on values between 0 and 1 (the larger the value, the stronger the association). The proteins come from 8 species.

**Prediction task**. The task is to predict the presence of protein functions in a multi-label binary classification setup, where there are 112 kinds of labels to predict in total. The performance is measured by the average of ROC-AUC scores across the 112 tasks.

**Dataset splitting**. We split the protein nodes into training/validation/test sets according to the species which the proteins come from. This enables the evaluation of the generalization performance of the model *across* different species.

**Discussion**. The `ogbn-proteins` dataset does not have input node features[10], but has edge features on more than 30 million edges. In our baseline experiments, we opt for simplicity and use the average edge features of incoming edges as node features.

Benchmarking results are shown in Table 8. Surprisingly, simple MLPs[11] perform better than more sophisticated approaches like NODE2VEC and GCN. Only GRAPHSAGE is able to outperform the naïve MLP approach, which indicates that central node information (that is not explicitly modeled in GCN in its message-passing) already contains a lot of crucial information for making correct predictions.

We further evaluate the best performing GRAPHSAGE on conventional *random split* with the same split ratio as the species split. On the random split, we find the generalization gap is extremely low, with $87.83_{\pm0.03}$% test ROC-AUC that is only 0.27 percentage points lower than the training ROC-AUC ($88.10_{\pm0.01}$%). This is in contrast to 10.18 percentage points of generalization gap (training AUC minus test AUC) in the species split, as calculated from the GRAPHSAGE experiment in Table 8. The result suggests the unique challenge of *across-species* generalization that needs to be tackled in future research.

Since the number of nodes in `ogbn-proteins` is fairly small and easily fit onto common GPUs, we did not run the CLUSTERGCN and GRAPHSAINT experiments. Nonetheless, this dataset presents an interesting research question of how to utilize edge features in a more sophisticated way than just naïve averaging, *e.g.*, by the usage of attention or by treating the graph as a multi-relational graph (as there are 8 different association types between proteins). The challenge is to scalably handle the huge number of edge features efficiently on GPU, which might require clever graph partitioning based on the edge weights.

## B.3 `ogbn-arxiv`: Paper Citation Network

The `ogbn-arxiv` dataset is a directed graph, representing the citation network between all Computer Science (CS) ARXIV papers indexed by MAG [87]. Each node is an ARXIV paper and each directed edge indicates that one paper cites another one. Each paper comes with a 128-dimensional

Table 8: **Results for `ogbn-proteins`.**

| Method | ROC-AUC (%) | | |
|---|---|---|---|
| | Training | Validation | Test |
| MLP | $81.78_{\pm 0.48}$ | $77.06_{\pm 0.14}$ | $72.04_{\pm 0.48}$ |
| NODE2VEC | $79.76_{\pm 1.88}$ | $70.07_{\pm 0.53}$ | $68.81_{\pm 0.65}$ |
| GCN | $82.77_{\pm 0.16}$ | $79.21_{\pm 0.18}$ | $72.51_{\pm 0.35}$ |
| GRAPHSAGE | $87.86_{\pm 0.13}$ | $\mathbf{83.34}_{\pm 0.13}$ | $\mathbf{77.68}_{\pm 0.20}$ |

Table 9: **Results for `ogbn-arxiv`.**

| Method | Accuracy (%) | | |
|---|---|---|---|
| | Training | Validation | Test |
| MLP | $63.58_{\pm 0.81}$ | $57.65_{\pm 0.12}$ | $55.50_{\pm 0.23}$ |
| NODE2VEC | $76.43_{\pm 0.81}$ | $71.29_{\pm 0.13}$ | $70.07_{\pm 0.13}$ |
| GCN | $78.87_{\pm 0.66}$ | $\mathbf{73.00}_{\pm 0.17}$ | $\mathbf{71.74}_{\pm 0.29}$ |
| GRAPHSAGE | $82.35_{\pm 1.51}$ | $72.77_{\pm 0.16}$ | $71.49_{\pm 0.27}$ |

feature vector obtained by averaging the embeddings of words in its title and abstract. The embeddings of individual words are computed by running the WORD2VEC model [62] over the MAG corpus. In addition, all papers are also associated with the year that the corresponding paper was published.

**Prediction task**. The task is to predict the 40 subject areas of ARXIV CS papers,[12] *e.g.*, *cs.AI*, *cs.LG*, and *cs.OS*, which are manually determined (*i.e.*, labeled) by the paper's authors and ARXIV moderators. With the volume of scientific publications doubling every 12 years over the past century [27], it is practically important to automatically classify each publication's areas and topics. Formally, the task is to predict the primary categories of the ARXIV papers, which is formulated as a 40-class classification problem.

**Dataset splitting**. The previously-used Cora, CiteSeer, and PubMed citation networks are split randomly [98]. In contrast, we consider a realistic data split based on the publication dates of the papers. The general setting is that the ML models are trained on existing papers and then used to predict the subject areas of newly-published papers, which supports the direct application of them into real-world scenarios, such as helping the ARXIV moderators. Specifically, we propose to train on papers published until 2017, validate on those published in 2018, and test on those published since 2019.

**Discussion**. Our initial benchmarking results are shown in Table 9, where the directed graph is converted to an undirected one for simplicity. First, we observe that the naïve MLP baseline that does not utilize any graph information is significantly outperformed by the other three models that utilize graph information. This suggests that graph information can dramatically improve the performance of predicting a paper's category. Comparing models that do utilize graph information, we find GNN models, *i.e.*, GCN and GRAPHSAGE, slightly outperform the NODE2VEC model. We also conduct additional experiments on conventional *random split* with the same split ratio. On the random split, we find that GCN achieves $73.54_{\pm 0.13}\%$ test accuracy, suggesting that the realistic time split is indeed more challenging than the random split, providing an opportunity to improve the out-of-distribution generalization performance. Furthermore, we think it will be fruitful to explore how the edge direction information as well as the node temporal information (e.g., year in which papers are published) can be taken into account to improve prediction performance.

### B.4 `ogbn-papers100M`: Paper Citation Network

The `ogbn-papers100M` dataset is a directed citation graph of 111 million papers indexed by MAG [87]. Its graph structure and node features are constructed in the same way as `ogbn-arxiv` in Appendix B.3. Among its node set, approximately 1.5 million of them are ARXIV papers, each of

Table 10: **Results for `ogbn-papers100M`.**

| Method | Accuracy (%) | | |
|--------|----------|------------|------|
|        | Training | Validation | **Test** |
| MLP    | $54.84_{\pm 0.43}$ | $49.60_{\pm 0.29}$ | $47.24_{\pm 0.31}$ |
| SGC    | $67.54_{\pm 0.43}$ | $\mathbf{66.48}_{\pm 0.20}$ | $\mathbf{63.29}_{\pm 0.19}$ |

which is manually labeled with one of ARXIV's subject areas (*cf.*Appendix B.3). Overall, this dataset is orders-of-magnitude larger than any existing node classification datasets.

**Prediction task**. Given the full `ogbn-papers100M` graph, the task is to predict the subject areas of the subset of papers that are published in ARXIV. The majority of nodes (corresponding to non-ARXIV papers) are not associated with label information, and only their node features and reference information are given. The task is to leverage the entire citation network to infer the labels of the ARXIV papers.[13] In total, there are 172 ARXIV subject areas, making the prediction task a 172-class classification problem.

**Dataset splitting**. The splitting strategy is the same as that used in `ogbn-arxiv`, *i.e.*, the time-based split. Specifically, the training nodes (with labels) are all ARXIV papers published until 2017, while the validation nodes are the ARXIV papers published in 2018, and the models are tested on ARXIV papers published since 2019.

**Discussion**. Our initial benchmarking results are shown in Table 10, where the directed graph is converted to an undirected one for simplicity. As most existing models have difficulty handling such a gigantic graph, we benchmark the two simplest models,[14] MLP and SGC [91], which is a simplified variant of textscGCN [49] that essentially pre-processes node features using graph adjacency information. We obtain SGC node embeddings on the CPU (requiring more than 100GB of memory), after which we train the final MLP with mini-batches on an ordinary GPU.

We see from Table 10 that the graph-based model, SGC, despite its simplicity, performs much better than the naïve MLP baseline. Nevertheless, we observe severe underfitting of SGC, indicating that using more expressive GNNs is likely to improve both training and test accuracy. It is therefore fruitful to explore how to scale expressive and advanced GNNs to the gigantic Web-scale graph, going beyond the simple pre-processing of node features. Overall, `ogbn-papers100M` is by far the largest benchmark dataset for node classification over a homogeneous graph, and is meant to significantly push the scalability of graph models.

### B.5 `ogbn-mag`: Heterogeneous Microsoft Academic Graph (MAG)

The `ogbn-mag` dataset is a heterogeneous network composed of a subset of the Microsoft Academic Graph (MAG) [87]. It contains four types of entities—papers (736,389 nodes), authors (1,134,649 nodes), institutions (8,740 nodes), and fields of study (59,965 nodes)—as well as four types of directed relations connecting two types of entities—an author "is affiliated with" an institution,[15] an author "writes" a paper, a paper "cites" a paper, and a paper "has a topic of" a field of study. Similar to `ogbn-arxiv` described in Appendix B.3, each paper is associated with a 128-dimensional WORD2VEC feature vector, and all the other types of entities are not associated with input node features.

**Prediction task**. Given the heterogeneous `ogbn-mag` data, the task is to predict the venue (conference or journal) of each paper, given its content, references, authors, and authors' affiliations. This is of practical interest as some manuscripts' venue information is unknown or missing in MAG, due to the noisy nature of Web data. In total, there are 349 different venues in `ogbn-mag`, making the task a 349-class classification problem.

**Dataset splitting**. We follow the same time-based strategy as `ogbn-arxiv` and `ogbn-papers100M` to split the paper nodes in the heterogeneous graph, *i.e.*, training models

Table 11: **Results for `ogbn-mag`.**
[†]Requires a GPU with 14GB of memory.

| Method | Accuracy (%) | | |
|--------|----------|------------|------|
| | Training | Validation | Test |
| MLP | $28.33_{\pm 0.20}$ | $26.26_{\pm 0.16}$ | $26.92_{\pm 0.26}$ |
| GCN | $29.71_{\pm 0.19}$ | $29.53_{\pm 0.22}$ | $30.43_{\pm 0.25}$ |
| GRAPHSAGE | $30.79_{\pm 0.19}$ | $30.70_{\pm 0.19}$ | $31.53_{\pm 0.15}$ |
| METAPATH2VEC | $38.35_{\pm 1.39}$ | $35.06_{\pm 0.17}$ | $35.44_{\pm 0.36}$ |
| R-GCN[†] | $75.87_{\pm 4.19}$ | $40.84_{\pm 0.41}$ | $39.77_{\pm 0.46}$ |
| NEIGHBORSAMPLING | $68.53_{\pm 7.27}$ | $47.61_{\pm 0.68}$ | $46.78_{\pm 0.67}$ |
| CLUSTERGCN | $79.65_{\pm 4.12}$ | $38.40_{\pm 0.31}$ | $37.32_{\pm 0.37}$ |
| GRAPHSAINT | $79.64_{\pm 1.70}$ | $\mathbf{48.37}_{\pm 0.26}$ | $\mathbf{47.51}_{\pm 0.22}$ |

to predict venue labels of all papers published before 2018, validating and testing the models on papers published in 2018 and since 2019, respectively.

**Discussion**. As `ogbn-mag` is a heterogeneous graph, we consider slightly different sets of GNN and node embedding baselines. Specifically, for GCN and GRAPHSAGE, as they are originally designed for homogeneous graphs, we apply the models over the homogeneous subgraph, retaining only paper nodes and their citation relations. We also consider the RELATIONAL-GCN (R-GCN) [72] that is specifically designed for heterogeneous graphs and uses specialized message passing parameters for different edge types. Since only "paper" nodes come with node features, we use trainable embeddings for the remaining nodes. For the node embedding model, instead of NODE2VEC, we adopt METAPATH2VEC [26], as it is specifically designed for heterogeneous graphs. For each relation, *e.g.*, an author "writes" a paper, the reverse relation, *e.g.*, a paper "is written by" an author, is added to allow bidirectional message passing in GNNs.

Our benchmarking results are shown in Table 11. First, we see that MLP, GCN, and GRAPHSAGE perform worse than the models that actually utilize heterogeneous graph information, *i.e.*, METAP-ATH2VEC, R-GCN, and the mini-batch GNNs.[16] This highlights that exploiting the heterogeneous nature of the graph is essential to achieving good performance on this dataset.

Second, we see that the mini-batch GNNs, especially NEIGHBORSAMPLING and GRAPHSAINT, give surprisingly promising results, outperforming the full-batch R-GCN by a large margin. This is likely due to the regularization effect of the noise induced by mini-batch sampling and edge dropout [71]. In contrast, CLUSTERGCN gives worse performance than its full-batch variant, indicating that the bias introduced by the pre-computed partitioning has a negative effect on the model's performance (as can be also seen by its highly overfitting training performance).

Nevertheless, heterogeneous graph models as well as their mini-batch training methods have been much less explored compared to their homogeneous counterparts, due to the smaller number of established benchmarks. Overall, `ogbn-mag` is meant to catalyze the development of scalable and accurate heterogeneous graph models, going beyond homogeneous graphs. A fruitful research direction is to adopt advanced techniques developed for homogeneous graphs to improve the performance on heterogeneous graphs.

## C  Comprehensive Descriptions of OGB Link Property Prediction

We currently provide 6 datasets, adopted from diverse application domains for predicting the properties of links (pairs of nodes). Specifically, `ogbl-ppa` is a protein-protein association network [80] (*cf.* Appendix C.1), `ogbl-collab` is an author collaboration network [87] (*cf.* Appendix C.2), `ogbl-ddi` is a drug-drug interaction network [90] (*cf.* Appendix C.3), `ogbl-citation` is a paper citation network [87] (*cf.* Appendix C.4), `ogbl-biokg` is a heterogeneous knowledge graph

compiled from a large number of biomedical repositories (*cf.* Appendix C.6), and `ogbl-wikikg` is a Wikidata knowledge graph [85] (*cf.* Appendix C.5).

The different datasets are highly diverse in their graph structure, as shown in Table 2. For example, the biological networks (`ogbl-ppa` and `ogbl-ddi`) are much denser than the academic networks (`ogbl-collab` and `ogbl-citation`) and the knowledge graphs (`ogbl-wikikg` and `ogbl-biokg`), as can be seen from the large average node degree, small number of nodes, and the small graph diameter. On the other hand, the collaboration network, `ogbl-collab`, has more clustered graph structure than the other datasets, as can be seen from its high average clustering coefficient. Comparing the two knowledge graph datasets, `ogbl-wikikg` and `ogbl-biokg`, we see that the former is much more sparse and less clustered than the latter.

**Baselines**. We implement different sets of baselines for link prediction datasets that only have a single edge type, and KG completion datasets that have multiple edge/relation types.

**Baselines for link prediction datasets**. We consider the following representative models as our baselines for the link prediction datasets unless otherwise specified. For all models, edge features are obtained by using the Hadamard operator $\odot$ between pair-wise node embeddings, and are then inputted to an MLP for the final prediction. During training, we randomly sample edges and use them as negative examples. We use the same number of negative edges as there are positive edges. Below, we describe how each model obtains node embeddings:

- **MLP**: Input node features are directly used as node embeddings.
- **NODE2VEC**: The node embeddings are obtained by concatenating input features and NODE2VEC embeddings [35, 66].
- **GCN**: The node embeddings are obtained by full-batch Graph Convolutional Networks (GCN) [49].
- **GRAPHSAGE**: The node embeddings are obtained by full-batch GraphSAGE [37], where we adopt its mean pooling variant and a simple skip connection to preserve central node features.
- **MATRIXFACTORIZATION**: The distinct embeddings are assigned to different nodes and are learned in an end-to-end manner together with the MLP predictor.
- **NEIGHBORSAMPLING** (optional): A mini-batch training technique of GNNs [37] that samples neighborhood nodes when performing aggregation.
- **CLUSTERGCN** (optional): A mini-batch training technique of GNNs [17] that partitions the graphs into a fixed number of subgraphs and draws mini-batches from them.
- **GRAPHSAINT** (optional): A mini-batch training technique of GNNs [103] that samples subgraphs via a random walk sampler.

Similar to the node property prediction baselines, the mini-batch GNN training, NEIGHBORSAMPLING, CLUSTERGCN, and GRAPHSAINT, are experimented only for graph datasets where full-batch GCN and GRAPHSAGE did not fit into the common GPU memory size of 11GB. To choose the GNN architecture for the mini-batch GNNs, we first run full-batch GCN and GRAPHSAGE on a NVIDIA Quadro RTX 8000 with 48GB of memory, and then adopt the best performing full-batch GNN architecture for the mini-batch GNNs. All models are trained with a fixed hidden dimensionality of 256, a fixed number of three layers, and a tuned dropout ratio $\in \{0.0, 0.5\}$.

**Baselines for KG completion datasets**. We consider the following representative KG embedding models as our baselines for the KG datasets unless otherwise specified.

- **TRANSE**: Translation-based KG embedding model by Bordes *et al.* [14].
- **DISTMULT**: Multiplication-based KG embedding model by Yang *et al.* [96].
- **COMPLEX**: Complex-valued multiplication-based KG embedding model by Trouillon *et al.* [81].
- **ROTATE**: Rotation-based KG embedding model by Sun *et al.* [78].

For KGs with many entities and relations, the embedding dimensionality can be limited by the available GPU memory, as the embeddings need to be loaded into GPU all at once. We therefore choose the dimensionality such that training can be performed on a fixed-budget of GPU memory. Our training procedure follows Sun *et al.* [78], where we perform negative sampling and use margin-based logistic loss for the loss function.

Table 12: **Results for `ogbl-ppa`.**

| Method | Hits@100 (%) | | |
|---|---|---|---|
| | Training | Validation | **Test** |
| MLP | $0.46_{\pm 0.00}$ | $0.46_{\pm 0.00}$ | $0.46_{\pm 0.00}$ |
| NODE2VEC | $24.43_{\pm 0.92}$ | $22.53_{\pm 0.88}$ | $22.26_{\pm 0.83}$ |
| GCN | $19.89_{\pm 1.51}$ | $18.45_{\pm 1.40}$ | $18.67_{\pm 1.32}$ |
| GRAPHSAGE | $18.53_{\pm 2.85}$ | $17.24_{\pm 2.64}$ | $16.55_{\pm 2.40}$ |
| MATRIXFACTORIZATION | $81.65_{\pm 9.15}$ | $\mathbf{32.28}_{\pm 4.28}$ | $\mathbf{32.29}_{\pm 0.94}$ |

## C.1 `ogbl-ppa`: Protein-Protein Association Network

The `ogbl-ppa` dataset is an undirected, unweighted graph. Nodes represent proteins from 58 different species, and edges indicate biologically meaningful associations between proteins, *e.g.*, physical interactions, co-expression, homology or genomic neighborhood [80]. We provide a graph object constructed from training edges (*i.e.*, no validation and test edges are contained). Each node contains a 58-dimensional one-hot feature vector that indicates the species that the corresponding protein comes from.

**Prediction task**. The task is to predict new association edges given the training edges. The evaluation is based on how well a model ranks positive test edges over negative test edges. Specifically, we rank each positive edge in the validation/test set against 3,000,000 randomly-sampled negative edges, and count the ratio of positive edges that are ranked at the $K$-th place or above (Hits@$K$). We found $K = 100$ to be a good threshold to rate a model's performance in our initial experiments. Overall, this metric is much more challenging than ROC-AUC because the model needs to consistently rank the positive edges higher than *nearly all* the negative edges.

**Dataset splitting**. We provide a biological throughput split of the edges into training/validation/test edges. Training edges are protein associations that are measured experimentally by a high-throughput technology (*e.g.*, cost-effective, automated experiments that make large scale repetition feasible [8, 60, 102]) or are obtained computationally (*e.g.*, via text-mining). In contrast, validation and test edges contain protein associations that can only be measured by low-throughput, resource-intensive experiments performed in laboratories. In particular, the goal is to predict a particular type of protein association, *e.g.*, physical protein-protein interaction, from other types of protein associations (*e.g.*, co-expression, homology, or genomic neighborhood) that can be more easily measured and are known to correlate with associations that we are interested in.

**Discussion**. Our initial benchmarking results are shown in Table 12. First, the MLP baseline[17] performs extremely poorly, which is to be expected since the node features are not rich in this dataset. Surprisingly, both GNN baselines (GCN, GRAPHSAGE) and NODE2VEC fail to overfit on the training data and show similar performances across training/validation/test splits. The poor training performance of GNNs suggests that *positional* information, which cannot be captured by GNNs alone [101], might be crucial to fit training edges and obtain meaningful node embeddings. On the other hand, we see that MATRIXFACTORIZATION, which learns a distinct embedding for each node (thus, it can express any positional information of nodes), is indeed able to overfit on the training data, while also reaching better validation and test performance. However, the poor generalization performance still encourages the development of new research ideas to close this gap, *e.g.*, by injecting positional information into GNNs or by developing more sophisticated negative sampling techniques.

## C.2 `ogbl-collab`: Author Collaboration Network

The `ogbl-collab` dataset is an undirected graph, representing a subset of the collaboration network between authors indexed by MAG [87]. Each node represents an author and edges indicate the collaboration between authors. All nodes come with 128-dimensional features, obtained by averaging the word embeddings of papers that are published by the authors. All edges are associated with two types of meta-information: the year and the edge weight, representing the number of co-authored papers published in that year. The graph can be viewed as a dynamic multi-graph since there can be multiple edges between two nodes if they collaborate in more than one year.

Table 13: **Results for `ogbl-collab`.**

| Method | Use most recent edges | Hits@50 (%) Training | Validation | Test |
|---|:---:|:---:|:---:|:---:|
| MLP | ✗ | $45.70_{\pm 1.66}$ | $24.02_{\pm 1.45}$ | $19.27_{\pm 1.29}$ |
| NODE2VEC | ✗ | $99.73_{\pm 0.36}$ | $\mathbf{57.03}_{\pm 0.52}$ | $\mathbf{48.88}_{\pm 0.54}$ |
| GCN | ✗ | $84.28_{\pm 1.78}$ | $52.63_{\pm 1.15}$ | $44.75_{\pm 1.07}$ |
| GRAPHSAGE | ✗ | $93.58_{\pm 0.59}$ | $56.88_{\pm 0.77}$ | $48.10_{\pm 0.81}$ |
| MATRIXFACTORIZATION | ✗ | $100.00_{\pm 0.00}$ | $48.96_{\pm 0.29}$ | $38.86_{\pm 0.29}$ |
| GCN | ✔ | $84.28_{\pm 1.78}$ | $52.63_{\pm 1.15}$ | $47.14_{\pm 1.45}$ |
| GRAPHSAGE | ✔ | $93.58_{\pm 0.59}$ | $56.88_{\pm 0.77}$ | $\mathbf{54.63}_{\pm 1.12}$ |

**Prediction task**. The task is to predict the author collaboration relationships in a particular year given the past collaborations. As the task is a time-series problem, it is natural for models to incorporate the most recent edge information to make prediction, *e.g.*, use validation edges when predicting test edges. The evaluation metric is similar to `ogbl-ppa` in Appendix C.1, where we would like the model to rank true collaborations higher than false collaborations. Specifically, we rank each true collaboration among a set of 100,000 randomly-sampled negative collaborations, and count the ratio of positive edges that are ranked at $K$-place or above (Hits@$K$). We found $K = 50$ to be a good threshold in our preliminary experiments.

**Dataset splitting**. We split the data according to time, in order to simulate a realistic application in collaboration recommendation. Specifically, we use the collaborations until 2017 as training edges, those in 2018 as validation edges, and those in 2019 as test edges.

**Discussion**. Our initial benchmarking results are shown in Table 13. First, we consider the conventional setting where validation edges are used only for model selection. From the upper half of Table 13, we see that NODE2VEC achieves the best results, followed by the two GNN models and MATRIXFACTORIZATION. This can be explained by the fact that positional information, *i.e.*, past collaborations, is a much more indicative feature for predicting future collaboration than solely relying on the average paper representations of authors, *i.e.*, the same research interest. Notably, MATRIXFACTORIZATION achieves nearly perfect training results, but cannot transfer the good results to the validation and test splits, even when heavy regularization is applied. Overall, it is fruitful to explore injecting positional information into GNNs, and develop better regularization methods. This dataset further provides a unique research opportunity for dynamic multi-graphs. To demonstrate the potential benefit of time-series modeling, we use the same GCN and GRAPHSAGE models as before but at test time, we additionally incorporate the most recent edges (*i.e.*, validation edges) as input to the models. From the lower half of Table 13, we see that the test performances of both GNN models increase significantly by using validation edges at the inference time. One promising direction to further increase the performance is to treat edges at different timestamps differently, as recent collaborations may be more indicative about the future collaborations than the past ones.

## C.3 `ogbl-ddi`: Drug-Drug Interaction Network

The `ogbl-ddi` dataset is a homogeneous, unweighted, undirected graph, representing the drug-drug interaction network [90]. Each node represents an FDA-approved or experimental drug. Edges represent interactions between drugs and can be interpreted as a phenomenon where the joint effect of taking the two drugs together is considerably different from the expected effect in which drugs act independently of each other.

**Prediction task**. The task is to predict drug-drug interactions given information on already known drug-drug interactions. The evaluation metric is similar to `ogbl-collab` discussed in Appendix C.2, where we would like the model to rank true drug interactions higher than non-interacting drug pairs. Specifically, we rank each true drug interaction among a set of approximately 100,000 randomly-sampled negative drug interactions, and count the ratio of positive edges that are ranked at $K$-place or above (Hits@$K$). We found $K = 20$ to be a good threshold in our preliminary experiments.

Table 14: **Results for `ogbl-ddi`.**

| Method | Hits@20 (%) | | |
|---|---|---|---|
| | Training | Validation | **Test** |
| NODE2VEC | $37.82_{\pm 1.35}$ | $32.92_{\pm 1.21}$ | $23.26_{\pm 2.09}$ |
| GCN | $63.95_{\pm 2.17}$ | $55.50_{\pm 2.08}$ | $37.07_{\pm 5.07}$ |
| GRAPHSAGE | $72.24_{\pm 0.45}$ | $\mathbf{62.62}_{\pm 0.37}$ | $\mathbf{53.90}_{\pm 4.74}$ |
| MATRIXFACTORIZATION | $56.56_{\pm 13.88}$ | $33.70_{\pm 2.64}$ | $13.68_{\pm 4.75}$ |

**Dataset splitting**. We develop a *protein-target split*, meaning that we split drug edges according to what proteins those drugs target in the body. As a result, the test set consists of drugs that predominantly bind to different proteins from drugs in the train and validation sets. This means that drugs in the test set work differently in the body, and have a rather different biological mechanism of action than drugs in the train and validation sets. The protein-target split thus enables us to evaluate to what extent the models can generate practically useful predictions [36], *i.e.*, non-trivial predictions that are not hindered by the assumption that there exist already known and very similar medications available for training.

**Discussion**. Our initial benchmarking results are shown in Table 14. Since `ogbl-ddi` does not contain any node features, we omit the graph-agnostic MLP baseline for this experiment. Furthermore, for GCN and GRAPHSAGE, node features are also represented as distinct embeddings and learned in an end-to-end manner together with the GNN parameters.

Interestingly, both the GNN models and the MATRIXFACTORIZATION approach achieve significantly higher training results than NODE2VEC. However, only the GNN models are able to transfer this performance to the test set to some extent, suggesting that relational information is crucial to allow the model to generalize to unseen interactions. Notably, most of the models show high performance variance, which can be partly attributed to the dense nature of the graph and the challenging data split. We further perform the conventional random split of edges, where we find GRAPHSAGE is able to achieve $80.88_{\pm 2.42}\%$ test Hits@20. This indicates that the protein-target split is indeed more challenging than the conventional random split. Overall, `ogbl-ddi` presents a unique challenge of predicting out-of-distribution links in dense graphs.

### C.4 `ogbl-citation`: Paper Citation Network

The `ogbl-citation` dataset is a directed graph, representing the citation network between a subset of papers extracted from MAG [87]. Similar to `ogbn-arxiv` in Appendix B.3, each node is a paper with 128-dimensional WORD2VEC features that summarizes its title and abstract, and each directed edge indicates that one paper cites another. All nodes also come with meta-information indicating the year the corresponding paper was published.

**Prediction task**. The task is to predict missing citations given existing citations. Specifically, for each source paper, two of its references are randomly dropped, and we would like the model to rank the missing two references higher than 1,000 negative reference candidates. The negative references are randomly-sampled from all the previous papers that are not referenced by the source paper. The evaluation metric is Mean Reciprocal Rank (MRR), where the reciprocal rank of the true reference among the negative candidates is calculated for each source paper, and then the average is taken over all source papers.

**Dataset splitting**. We split the edges according to time, in order to simulate a realistic application in citation recommendation (*e.g.*, a user is writing a new paper and has already cited several existing papers, but wants to be recommended additional references). To this end, we use the most recent papers (those published in 2019) as the source papers for which we want to recommend the references. For each source paper, we drop *two* papers from its references—the resulting two dropped edges (pointing from the source paper to the dropped papers) are used respectively for validation and testing. All the rest of the edges are used for training.

**Discussion**. Our initial benchmarking results are shown in Table 15, where the directed graph is converted to an undirected one for simplicity. Here, the GNN models achieve the best results,

Table 15: **Results for `ogbl-citation`.**
[†]Requires a GPU with 40GB of memory

| Method | MRR | | |
| --- | --- | --- | --- |
| | Training | Validation | **Test** |
| MLP | $0.2889_{\pm0.0014}$ | $0.2898_{\pm0.0014}$ | $0.2904_{\pm0.0013}$ |
| NODE2VEC | $0.6831_{\pm0.0011}$ | $0.5944_{\pm0.0011}$ | $0.5964_{\pm0.0011}$ |
| GCN[†] | $0.9064_{\pm0.0100}$ | $\mathbf{0.8449}_{\pm0.0108}$ | $\mathbf{0.8456}_{\pm0.0110}$ |
| GRAPHSAGE[†] | $0.8891_{\pm0.0079}$ | $0.8217_{\pm0.0086}$ | $0.8228_{\pm0.0084}$ |
| MATRIXFACTORIZATION | $0.9171_{\pm0.0179}$ | $0.5311_{\pm0.0565}$ | $0.5316_{\pm0.0565}$ |
| NEIGHBORSAMPLING | $0.8621_{\pm0.0008}$ | $0.8048_{\pm0.0013}$ | $0.8048_{\pm0.0015}$ |
| CLUSTERGCN | $0.8754_{\pm0.0033}$ | $0.7999_{\pm0.0027}$ | $0.8021_{\pm0.0029}$ |
| GRAPHSAINT | $0.8626_{\pm0.0046}$ | $0.7933_{\pm0.0046}$ | $0.7943_{\pm0.0043}$ |

followed by MATRIXFACTORIZATION and NODE2VEC. Among the GNNs, GCN performs better than GRAPHSAGE. However, these GNNs use full-batch training; thus, they are not scalable and require more than 40GB of GPU memory to train, which is intractable on most of the GPUs available today. Hence, we also experiment with the scalable mini-batch training techniques of GNNs, NEIGHBORSAMPLING, CLUSTERGCN, and GRAPHSAINT. Interestingly, we see from Table 15 that these techniques give worse performance than their full-batch counterpart, which is in contrast to the node classification datasets (*e.g.*, `ogbn-products` and `ogbn-mag`), where the mini-batch-based models give stronger generalization performances. This limitation presents a unique challenge for applying the mini-batch techniques to link prediction, differently from those pertaining to node prediction. Overall, `ogbl-citation` provides a research opportunity to further improve GNN models and their scalable mini-batch training techniques in the context of link prediction.

### C.5 `ogbl-wikikg`: Wikidata Knowledge Graph

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

## C.6 `ogbl-biokg`: Biomedical Knowledge Graph

The `ogbl-biokg` dataset is a Knowledge Graph (KG), which we created using data from a large number of biomedical data repositories. It contains 5 types of entities: diseases (10,687 nodes), proteins (17,499), drugs (10,533 nodes), side effects (9,969 nodes), and protein functions (45,085 nodes). There are 51 types of directed relations connecting two types of entities, including 39 kinds of drug-drug interactions, 8 kinds of protein-protein interaction, as well as drug-protein, drug-side effect, drug-protein, function-function relations. All relations are modeled as directed edges, among which the relations connecting the same entity types (*e.g.*, protein-protein, drug-drug, function-function) are always symmetric, *i.e.*, the edges are bi-directional.

Table 17: **Results for `ogbl-biokg`.**

| Method | MRR | | |
|--------|-----|---|---|
| | Training (Unfiltered) | Validation (Filtered) | **Test** (Filtered) |
| TRANSE | $0.5145_{\pm 0.0005}$ | $0.7456_{\pm 0.0003}$ | $0.7452_{\pm 0.0004}$ |
| DISTMULT | $0.5250_{\pm 0.0006}$ | $0.8055_{\pm 0.0003}$ | $0.8043_{\pm 0.0003}$ |
| COMPLEX | $0.5315_{\pm 0.0006}$ | $\mathbf{0.8105_{\pm 0.0001}}$ | $\mathbf{0.8095_{\pm 0.0007}}$ |
| ROTATE | $0.5363_{\pm 0.0007}$ | $0.7997_{\pm 0.0002}$ | $0.7989_{\pm 0.0004}$ |

This dataset is relevant to both biomedical and fundamental ML research. On the biomedical side, the dataset allows us to get better insights into human biology and generate predictions that can guide downstream biomedical research. On the fundamental ML side, the dataset presents challenges in handling a noisy, incomplete KG with possible contradictory observations. This is because the `ogbl-biokg` dataset involves heterogeneous interactions that span from the molecular scale (*e.g.*, protein-protein interactions within a cell) to whole populations (*e.g.*, reports of unwanted side effects experienced by patients in a particular country). Further, triplets in the KG come from sources with a variety of confidence levels, including experimental readouts, human-curated annotations, and automatically extracted metadata.

**Prediction task**. The task is to predict new triplets given the training triplets. The evaluation protocol is exactly the same as `ogbl-wikikg` in Appendix C.5, except that here we only consider ranking against entities *of the same type*. For instance, when corrupting head entities of the protein type, we only consider negative protein entities.

**Dataset splitting**. For this dataset, we adopt a random split. While splitting the triplets according to time is an attractive alternative, we note that it is incredibly challenging to obtain accurate information as to when individual experiments and observations underlying the triplets were made. We strive to provide additional dataset splits in future versions of the OGB.

**Discussion**. Our benchmark results are provided in Table 17, where we adopt 2000-dimensional embeddings for DISTMULT and TRANSE, and 1000-dimensional embeddings for the real and imaginary parts of ROTATE and COMPLEX. Negative sampling is performed only over entities of the same types. Similar to Table 16 in Appendix C.5, training MRR in Table 17 is an *unfiltered* metric.[22]

Among the four models, COMPLEX achieves the best test MRR, while TRANSE gives significantly worse performance compared to the other models. The worse performance of TRANSE can be explained by the fact that TRANSE cannot model symmetric relations [81] that are prevalent in this dataset, *e.g.*, protein-protein and drug-drug relations are all symmetric. Overall, it is of great practical interest to further improve the model performance. A promising direction is to develop a more specialized method for the *heterogeneous* knowledge graph, where multiple node types exist and the entire graph follows the pre-defined schema.

# D Comprehensive Descriptions of OGB Graph Property Prediction

We currently provide 4 datasets, adopted from 3 distinct application domains, for predicting the properties of entire graphs or subgraphs. Specifically, `ogbg-molhiv` and `ogbg-molpcba` are molecular graphs originally curated by Wu *et al.* [92] (*cf.* Appendix D.1), `ogbg-ppa` is a set of protein-protein association subgraphs [108] (*cf.* Appendix D.2), and `ogbg-code` is a collection of ASTs of source code [43] (*cf.* Appendix D.3).

The different datasets are highly diverse in their graph structure, as shown in Table 2. For example, compared with the other graph datasets, the biological subgraphs, `ogbg-ppa`, have much larger number of nodes per graph, as well as much denser and clustered graph structure, as seen by the large average node degree, large average clustering coefficient, and large graph diameter.

This is contrast to the molecular graphs, `ogbg-molhiv` and `ogbg-molpcba`, as well as the ASTs, `ogbg-code`, both of which are tree-like graphs—in fact, ASTs are exactly trees—with small average node degrees, small average clustering coefficient, and large average graph diameter. Despite the similarity, the molecular graphs and the ASTs are distinct in that the ASTs have much larger number of nodes with well-defined root nodes.

**Baselines**. We consider the following representative GNNs as our baselines unless otherwise specified. GNNs are used to obtain node embeddings, which are then pooled to give the embedding of the entire graph. Finally, a linear model is applied to the graph embedding to make predictions.

- **GCN**: Graph Convolutioanl Networks [48].
- **GCN+VIRTUAL NODE**: GCN that performs message passing over augmented graphs with virtual nodes, *i.e.*, additional nodes that are connected to all nodes in the original graphs [34, 44, 55].
- **GIN**: Graph Isomorphism Network [94].
- **GIN+VIRTUAL NODE**: GIN that performs message passing over augmented graphs with virtual nodes.

To include edge features, we follow Hu *et al.* [40] and add transformed edge features into the incoming node features. For all the experiments, we use 5-layer GNNs, average graph pooling, a hidden dimensionality of 300, and a tuned dropout ratio $\in \{0.0, 0.5\}$.

## D.1 `ogbg-mol*`: Molecular Graphs

The `ogbg-molhiv` and `ogbg-molpcba` datasets are two molecular property prediction datasets of different sizes: `ogbg-molhiv` (small) and `ogbg-molpcba` (medium). They are adopted from the MOLECULENET [92], and are among the largest of the MOLECULENET datasets. Besides the two main molecule datasets, we also provide the 10 other MOLECULENET datasets, which are summarized and benchmarked in Appendix F. These datasets can be used to stress-test molecule-specific methods [46, 97] and transfer learning [40]. All the molecules are pre-processed using RDKIT [51]. Each graph represents a molecule, where nodes are atoms, and edges are chemical bonds. Input node features are 9-dimensional, containing atomic number and chirality, as well as other *additional* atom features such as formal charge and whether the atom is in the ring. Input edge features are 3-dimensional, containing bond type, bond stereochemistry as well as an *additional* bond feature indicating whether the bond is conjugated. Note that the above additional features are not needed to uniquely identify molecules, and are not adopted in the previous work [40, 44]. In the experiments, we perform an ablation study on the molecule features and find that including the additional features improves generalization performance.

**Prediction task**. The task is to predict the target molecular properties as accurately as possible, where the molecular properties are cast as binary labels, *e.g.*, whether a molecule inhibits HIV virus replication or not. For evaluation metric, we closely follow Wu *et al.* [92]. Specifically, for `ogbg-molhiv`, we use ROC-AUC for evaluation. For `ogbg-molpcba`, as the class balance is extremely skewed (only 1.4% of data is positive) and the dataset contains multiple classification tasks, we use the Average Precision (AP) averaged over the tasks as the evaluation metric.[23]

**Dataset splitting**. We adopt the *scaffold splitting* procedure that splits the molecules based on their two-dimensional structural frameworks. The scaffold splitting attempts to separate structurally different molecules into different subsets, which provides a more realistic estimate of model performance in prospective experimental settings. The scaffold splitting was originally proposed by Wu *et al.* [92] and has been adopted by the follow-up works [40, 44, 70, 97]; however, the precise implementation differs significantly across works, making their results not directly comparable to each other. In OGB, we aim to standardize the scaffold split by adopting its most challenging version where test molecules are maximally diverse.

**Discussion**. Benchmarking results are given in Tables 18 and 19. We see that GIN with the additional features and VIRTUAL NODES provides the best performance in the two datasets. In Appendix F, we show that even for the other MOLECULENET datasets, the additional features consistently improve

Table 18: **Results for `ogbg-molhiv`.**

| Method | Additional Features | Virtual Node | ROC-AUC (%) | | |
|---|---|---|---|---|---|
| | | | Training | Validation | Test |
| GCN | ✗ | ✔ | $88.65_{\pm 1.01}$ | $83.73_{\pm 0.78}$ | $74.18_{\pm 1.22}$ |
| | ✔ | ✗ | $88.65_{\pm 2.19}$ | $82.04_{\pm 1.41}$ | $76.06_{\pm 0.97}$ |
| | ✔ | ✔ | $90.07_{\pm 4.69}$ | $83.84_{\pm 0.91}$ | $75.99_{\pm 1.19}$ |
| GIN | ✗ | ✔ | $93.89_{\pm 2.96}$ | $84.1_{\pm 1.05}$ | $75.2_{\pm 1.30}$ |
| | ✔ | ✗ | $88.64_{\pm 2.54}$ | $82.32_{\pm 0.90}$ | $75.58_{\pm 1.40}$ |
| | ✔ | ✔ | $92.73_{\pm 3.80}$ | $\mathbf{84.79}_{\pm 0.68}$ | $\mathbf{77.07}_{\pm 1.49}$ |

Table 19: **Results for `ogbg-molpcba`.**

| Method | Additional Features | Virtual Node | AP (%) | | |
|---|---|---|---|---|---|
| | | | Training | Validation | Test |
| GCN | ✗ | ✔ | $36.25_{\pm 0.71}$ | $23.88_{\pm 0.22}$ | $22.91_{\pm 0.37}$ |
| | ✔ | ✗ | $28.04_{\pm 0.58}$ | $20.59_{\pm 0.33}$ | $20.20_{\pm 0.24}$ |
| | ✔ | ✔ | $38.25_{\pm 0.50}$ | $24.95_{\pm 0.42}$ | $24.24_{\pm 0.34}$ |
| GIN | ✗ | ✔ | $45.70_{\pm 0.61}$ | $27.54_{\pm 0.25}$ | $26.61_{\pm 0.17}$ |
| | ✔ | ✗ | $37.05_{\pm 0.31}$ | $23.05_{\pm 0.27}$ | $22.66_{\pm 0.28}$ |
| | ✔ | ✔ | $46.96_{\pm 0.57}$ | $\mathbf{27.98}_{\pm 0.25}$ | $\mathbf{27.03}_{\pm 0.23}$ |

generalization performance. In OGB, we therefore include the additional node/edge features in our molecular graphs.

We further report the performance on the random splitting, keeping the split ratio the same as the scaffold splitting. We find the random split to be much easier than scaffold split. On random splits of `ogbg-molhiv` and `ogbg-molpcba`, the best GIN achieves the ROC-AUC of $82.73_{\pm 2.02}\%$ (5.66 percentage points higher than scaffold) and AP of $34.40_{\pm 0.90}\%$ (7.37 percentage points higher than scaffold), respectively. The same trend holds true for the other MOLECULENET datasets, *e.g.*, the best GIN performance on the random split of `ogbg-moltox21` is $86.03_{\pm 1.37}\%$, which is 8.46 percentage points higher than that of the best GIN for the scaffold split ($77.57_{\pm 0.62}\%$ ROC-AUC). These results highlight the challenges of the scaffold split compared to the random split, and opens up a fruitful research opportunity to increase the out-of-distribution generalization capability of GNNs.

## D.2 `ogbg-ppa`: Protein-Protein Association Network

The `ogbg-ppa` dataset is a set of undirected protein association neighborhoods extracted from the protein-protein association networks of 1,581 different species [80] that cover 37 broad taxonomic groups (*e.g.*, mammals, bacterial families, archaeans) and span the tree of life [42]. To construct the neighborhoods, we randomly selected 100 proteins from each species and constructed 2-hop protein association neighborhoods centered on each of the selected proteins [108]. We then removed the center node from each neighborhood and subsampled the neighborhood to ensure the final protein association graph is small enough (less than 300 nodes). Nodes in each protein association graph represent proteins, and edges indicate biologically meaningful associations between proteins. The edges are associated with 7-dimensional features, where each element takes a value between 0 and 1 and represents the strength of a particular type of protein protein association such as gene co-occurrence, gene fusion events, and co-expression.

**Prediction task**. Given a protein association neighborhood graph, the task is a 37-way multi-class classification to predict what taxonomic group the graph originates from. The ability to successfully tackle this problem has implications for understanding the evolution of protein complexes across species [22], the rewiring of protein interactions over time [73, 108], the discovery of functional

Table 20: **Results for `ogbg-ppa`.**

| Method | Virtual Node | Accuracy (%) | | |
|---|---|---|---|---|
| | | Training | Validation | Test |
| GCN | ✗ | $97.68_{\pm0.22}$ | $64.97_{\pm0.34}$ | $68.39_{\pm0.84}$ |
| | ✔ | $97.00_{\pm1.00}$ | $65.11_{\pm0.48}$ | $68.57_{\pm0.61}$ |
| GIN | ✗ | $97.55_{\pm0.52}$ | $65.62_{\pm1.07}$ | $68.92_{\pm1.00}$ |
| | ✔ | $98.28_{\pm0.46}$ | $\mathbf{66.78}_{\pm1.05}$ | $\mathbf{70.37}_{\pm1.07}$ |

Figure 5: Example input graph in `ogbg-code`, obtained by augmenting the original Python AST.

associations between genes even for otherwise rarely-studied organisms [19], and would give us insights into key bioinformatics tasks, such as biological network alignment [61].

**Dataset splitting**. Similar to `ogbn-proteins` in Appendix B.2, we adopt the *species split*, where the neighborhood graphs in validation and test sets are extracted from protein association networks of species that are *not* seen during training but belong to one of the 37 taxonomic groups. This split stress-tests the model's capability to extract graph features that are essential to the prediction of the taxonomic groups, which is important for biological understanding of protein associations.

**Discussion**. Benchmarking results are given in Table 20. Interestingly, similar to the `ogbg-mol*` datasets, GIN with VIRTUAL NODE provides the best performance. Nevertheless, the generalization gap is huge (almost 30 percentage points). For reference, we also experiment with the random splitting scenario, where we use the same model (GIN+VIRTUAL NODE) on the same split ratio. On the random split, the test accuracy is $92.91_{\pm0.27}\%$, which is more than 20 percentage points higher than the species split. This again encourages future research to improve the out-of-distribution generalization with more challenging and meaningful split procedure.

### D.3 `ogbg-code`: Abstract Syntax Tree of Source Code

The `ogbg-code` dataset is a collection of Abstract Syntax Trees (ASTs) obtained from approximately 450 thousands Python method definitions. Methods are extracted from a total of 13,587 different repositories across the most popular projects on GITHUB (where "popularity" is defined as number of stars and forks). Our collection of Python methods originates from GITHUB CodeSearch-Net [43] [24], a collection of datasets and benchmarks for machine-learning-based code retrieval. The authors paid particular attention to avoid common shortcomings of previous source code datasets [2], such as duplication of code and labels, low number of projects, random splitting, etc. In `ogbg-code`, we contribute an additional feature extraction step, which includes: AST edges, AST nodes (associated with features such as their types and attributes), tokenized method name. Altogether, `ogbg-code` allows us to capture source code with its underlying graph structure, beyond its token sequence representation.

**Prediction task**. The task is to predict the sub-tokens forming the method name, given the Python

Table 21: **Results for `ogbg-code`.**

| Method | Virtual Node | F1 score (%) | | |
|---|---|---|---|---|
| | | Training | Validation | Test |
| GCN | ✗ | $44.81_{\pm2.79}$ | $29.73_{\pm0.14}$ | $31.63_{\pm0.18}$ |
| | ✔ | $45.76_{\pm2.28}$ | $\mathbf{30.62}_{\pm0.07}$ | $\mathbf{32.63}_{\pm0.13}$ |
| GIN | ✗ | $44.97_{\pm3.91}$ | $29.81_{\pm0.14}$ | $31.63_{\pm0.20}$ |
| | ✔ | $47.01_{\pm2.10}$ | $30.20_{\pm0.16}$ | $32.04_{\pm0.18}$ |

method body represented by AST and its node features—*i.e.*, node type (from a pool of 97 types), node attributes (such as variable names, with a vocabulary size of 10002), depth in the AST, pre-order traversal index (as illustrated in Figure 5). This task is often referred to in the literature as "code summarization" [3, 6, 7], because the model is trained to find succinct and precise description (*i.e.*, the method name chosen by the developer) for a complete logical unit (*i.e.*, the method body). Code summarization is a representative task in the field of machine learning for code not only for its straightforward adoption in developer tools, but also because it is a proxy measure for assessing how well a model captures the code semantic [5]. Following Alon *et al.* [6, 7], we use an F1 score to evaluate predicted sub-tokens against ground-truth sub-tokens.[25] The average length of a method name in the ground-truth is 2.6 sub-tokens, following a power-law distribution.

**Dataset splitting**. We adopt a *project split* [2], where the ASTs for the train set are obtained from GITHUB projects that do not appear in the validation and test sets. This split respects the practical scenario of training a model on a large collection of source code (obtained, for instance, from the popular GITHUB projects), and then using it to predict method names on a separate code base. The project split stress-tests the model's ability to capture code's semantics, and avoids a model that trivially memorizes the idiosyncrasies of training projects (such as the naming conventions and the coding style of a specific developer) to achieve a high test score.

**Discussion**. Benchmarking results are given in Table 21, where we add "next-token edges" on top of the AST (as illustrated in Figure 5) to better capture the semantics of code graphs [25]. [26] For the decoder, we use independent linear classifiers to predict sub-tokens at each position of the sub-token sequence.[27] The evaluation is performed against the ground-truth sub-tokens. We see from Table 21 that GCN with VIRTUAL NODES provides the best performance. Nevertheless, we observe a huge generalization gap (more than 10 percentage points). For reference, we also experiment with the random splitting scenario, where we apply the same model (GCN+VIRTUAL NODE) on the same split ratio. On the random split, the test F1 score is $36.58_{\pm0.30}$%, which is approximately 4 percentage points higher than that of the project split in Table 21, indicating that the project split is indeed harder than the random split. Overall, this dataset presents an interesting research opportunity to improve out-of-distribution generalization under the meaningful project split, with a number of fruitful future directions: how to leverage the fact that the original graphs are actually trees with well-defined root nodes, how to pre-train GNNs to improve generalization [40], and how to design a better encoder-decoder architecture with the graph data. To facilitate these directions, we provide enough meta-information, such as the original code snippet as well as an easy-to-use script to transform raw Python code snippets into the ASTs.

# E OGB Package

The OGB package is designed to make the pipeline of Figure 2 easily accessible to researchers, by automating the data loading and the evaluation parts. OGB is fully compatible with PYTORCH and its associated graph libraries: PYTORCH GEOMETRIC and DEEP GRAPH LIBRARY. OGB additionally provides library-agnostic dataset objects that can be used by any other Python deep learning frameworks such as TENSORFLOW [1] and MXNET [16]. Below, we explain the data loading (*cf.* Appendix E.1) and evaluation (*cf.* Appendix E.2). For simplicity, we focus on the task of the graph property prediction (*cf.* Appendix 5) using PYTORCH GEOMETRIC. For the other tasks, libraries, and more details, refer to `https://ogb.stanford.edu`.

## E.1 OGB Data Loaders

The OGB package makes it easy to obtain a dataset object that is fully compatible with PYTORCH GEOMETRIC. As shown in Code Snippet 1, it can be done with only a single line of code, with the end-users only needing to specify the name of the dataset. The OGB package will then download, process, store, and return the requested dataset object. Furthermore, the standardized dataset splitting can be readily obtained from the dataset object.

```
>>> from ogb.graphproppred import PygGraphPropPredDataset
>>> dataset = PygGraphPropPredDataset(name="ogbg-molpcba")
# Pytorch Geometric dataset object
>>> split_idx = dataset.get_idx_split()
# Dictionary containing train/valid/test indices.
>>> train_idx = split_idx["train"]
# torch.tensor storing a list of training indices.
```

Code Snippet 1: **OGB Data Loader**

## E.2 OGB Evaluators

OGB also enables standardized and reliable evaluation with the `ogb.*.Evaluator` class. As shown in Code Snippet 2, the end-users first specify the dataset they want to evaluate their models on, after which the users can learn the format of the input they need to pass to the `Evaluator` object. The input format is dataset-dependent. For example, for the `ogbg-molpcba` dataset, the `Evaluator` object requires as input a dictionary with `y_true` (a matrix storing the ground-truth binary labels[28]), and `y_pred` (a matrix storing the scores output by the model). Once the end-users pass the specified dictionary as input, the `Evaluator` object returns the model performance that is appropriate for the dataset at hand, *e.g.*, the Average Precision for `ogbg-molpcba`.

```
>>> from ogb.graphproppred import Evaluator
# Get Evaluator for ogbg-molpcba
>>> evaluator = Evaluator(name = "ogbg-molpcba")
# Learn about the specification of input to the Evaluator.
>>> print(evaluator.expected_input_format)
# Prepare input that follows input spec.
>>> input_dict = {"y_true": y_true, "y_pred": y_pred}
# Get the model performance.
result_dict = evaluator.eval(input_dict)
```

Code Snippet 2: **OGB Evaluator**

Table 22: **Summary of `ogbg-mol*` datasets.** For all the datasets, we use the scaffold split with the split ratio of 80/10/10.

| Category | Name | #Graphs | Average #Nodes | Average #Edges | #Tasks | Task Type | Metric |
|---|---|---|---|---|---|---|---|
| **Molecular Graph** `ogbg-mol` | tox21 | 7,831 | 18.6 | 19.3 | 12 | Binary class. | ROC-AUC |
| | toxcast | 8,576 | 18.8 | 19.3 | 617 | Binary class. | ROC-AUC |
| | muv | 93,087 | 24.2 | 26.3 | 17 | Binary class. | AP |
| | bace | 1,513 | 34.1 | 36.9 | 1 | Binary class. | ROC-AUC |
| | bbbp | 2,039 | 24.1 | 26.0 | 1 | Binary class. | ROC-AUC |
| | clintox | 1,477 | 26.2 | 27.9 | 2 | Binary class. | ROC-AUC |
| | sider | 1,427 | 33.6 | 35.4 | 27 | Binary class. | ROC-AUC |
| | esol | 1,128 | 13.3 | 13.7 | 1 | Regression | RMSE |
| | freesolv | 642 | 8.7 | 8.4 | 1 | Regression | RMSE |
| | lipo | 4,200 | 27.0 | 29.5 | 1 | Regression | RMSE |

# F More Benchmark Results on `ogbg-mol*` Datasets

Here we perform benchmark experiments on the other 10 datasets from MOLECULENET [92]. The datasets are summarized in Table 22. The detailed description of each dataset is provided in Wu *et al.* [92]. We use the same experimental protocol and hyper-parameters as in Appendix D.1. The dropout rate is fixed to 0.5. As evaluation metrics, we adopt ROC-AUC for all the binary classification datasets except for `ogbg-molmuv` that exhibits significant class imbalance (only 0.2% of labels are positive). For the `ogbg-molmuv` dataset, we use Average Precision (AP), which is a more appropriate metric for heavily-imbalanced data [21, 92]. For the regression datasets, we adopt Root Mean Squared Error (RMSE); the lower, the better.

The benchmark results for each dataset are provided in Tables 23–32. We observe the followings.

- The additional features almost always help improve generalization performance. In fact, on top of GIN+VIRTUAL NODE, including the additional features gives either comparable or improved performance on 9 out of the 10 datasets (except for `ogbg-molbace` in Table 26). This motivates us to include these additional features in our OGB molecular graphs.

- Adding VIRTUAL NODES often improves generalization performance; for example, on top of GIN, adding VIRTUAL NODES gives either comparable or improved performance on 9 out of the 10 datasets (except for `ogbg-clintox` in Table 28).

- The optimal GNN architectures (GCN or GIN) vary across the datasets. This raises a natural question: can we design a GNN architecture that performs well *across* the molecule datasets?

Altogether, we hope our extensive benchmark results on a variety of molecule datasets provide useful baselines for further research on molecule-specific graph ML models.

Table 23: **Results for `ogbg-moltox21`.**

| Method | Add. Feat. | Virt. Node | ROC-AUC (%) Training | Validation | Test |
|---|---|---|---|---|---|
| GCN | ✗ | ✓ | $90.01_{\pm1.81}$ | $81.12_{\pm0.37}$ | $75.51_{\pm1.00}$ |
|  | ✓ | ✗ | $92.06_{\pm0.93}$ | $79.04_{\pm0.19}$ | $75.29_{\pm0.69}$ |
|  | ✓ | ✓ | $93.28_{\pm2.18}$ | $\mathbf{82.05}_{\pm0.43}$ | $\mathbf{77.46}_{\pm0.86}$ |
| GIN | ✗ | ✓ | $93.13_{\pm0.94}$ | $81.47_{\pm0.3}$ | $76.21_{\pm0.82}$ |
|  | ✓ | ✗ | $93.06_{\pm0.88}$ | $78.32_{\pm0.48}$ | $74.91_{\pm0.51}$ |
|  | ✓ | ✓ | $93.67_{\pm1.03}$ | $\mathbf{82.17}_{\pm0.35}$ | $\mathbf{77.57}_{\pm0.62}$ |

Table 24: **Results for `ogbg-moltoxcast`.**

| Method | Add. Feat. | Virt. Node | ROC-AUC (%) Training | Validation | Test |
|---|---|---|---|---|---|
| GCN | ✗ | ✓ | $88.89_{\pm0.88}$ | $70.52_{\pm0.34}$ | $66.33_{\pm0.35}$ |
|  | ✓ | ✗ | $85.21_{\pm1.69}$ | $67.48_{\pm0.33}$ | $63.54_{\pm0.42}$ |
|  | ✓ | ✓ | $89.89_{\pm0.8}$ | $71.65_{\pm0.38}$ | $\mathbf{66.71}_{\pm0.45}$ |
| GIN | ✗ | ✓ | $85.51_{\pm0.59}$ | $69.62_{\pm0.66}$ | $66.18_{\pm0.68}$ |
|  | ✓ | ✗ | $84.65_{\pm1.56}$ | $68.62_{\pm0.63}$ | $63.41_{\pm0.74}$ |
|  | ✓ | ✓ | $86.42_{\pm0.49}$ | $\mathbf{72.32}_{\pm0.35}$ | $66.13_{\pm0.50}$ |

Table 25: **Results for `ogbg-molmuv`.**

| Method | Add. Feat. | Virt. Node | AP (%) Training | Validation | Test |
|---|---|---|---|---|---|
| GCN | ✗ | ✓ | $6.67_{\pm3.87}$ | $8.48_{\pm1.58}$ | $2.48_{\pm2.83}$ |
|  | ✓ | ✗ | $22.72_{\pm6.9}$ | $21.4_{\pm1.46}$ | $\mathbf{11.39}_{\pm2.87}$ |
|  | ✓ | ✓ | $23.64_{\pm7.12}$ | $\mathbf{22.1}_{\pm1.98}$ | $10.98_{\pm2.91}$ |
| GIN | ✗ | ✓ | $26.49_{\pm7.24}$ | $15.74_{\pm2.19}$ | $7.91_{\pm2.13}$ |
|  | ✓ | ✗ | $17.94_{\pm4.06}$ | $19.00_{\pm2.15}$ | $8.78_{\pm2.07}$ |
|  | ✓ | ✓ | $25.95_{\pm7.85}$ | $17.42_{\pm1.32}$ | $9.84_{\pm2.71}$ |

Table 26: **Results for `ogbg-molbace`.**

| Method | Add. Feat. | Virt. Node | ROC-AUC (%) Training | Validation | Test |
|---|---|---|---|---|---|
| GCN | ✗ | ✓ | $87.85_{\pm5.08}$ | $78.99_{\pm2.03}$ | $71.44_{\pm4.01}$ |
|  | ✓ | ✗ | $91.74_{\pm1.90}$ | $73.74_{\pm1.49}$ | $\mathbf{79.15}_{\pm1.44}$ |
|  | ✓ | ✓ | $91.16_{\pm2.86}$ | $80.25_{\pm1.43}$ | $68.93_{\pm6.95}$ |
| GIN | ✗ | ✓ | $87.84_{\pm1.75}$ | $77.21_{\pm1.01}$ | $76.41_{\pm2.68}$ |
|  | ✓ | ✗ | $92.07_{\pm2.62}$ | $73.30_{\pm1.95}$ | $72.97_{\pm4.00}$ |
|  | ✓ | ✓ | $92.04_{\pm5.77}$ | $\mathbf{80.81}_{\pm1.71}$ | $73.46_{\pm5.24}$ |

Table 27: **Results for `ogbg-molbbbp`.**

| Method | Add. Feat. | Virt. Node | ROC-AUC (%) Training | Validation | Test |
|---|---|---|---|---|---|
| GCN | ✗ | ✓ | $90.42_{\pm3.82}$ | $93.46_{\pm0.27}$ | $68.62_{\pm2.19}$ |
|  | ✓ | ✗ | $96.97_{\pm1.31}$ | $94.74_{\pm0.31}$ | $68.87_{\pm1.51}$ |
|  | ✓ | ✓ | $98.29_{\pm1.79}$ | $\mathbf{95.95}_{\pm0.40}$ | $67.80_{\pm2.35}$ |
| GIN | ✗ | ✓ | $94.06_{\pm1.85}$ | $94.66_{\pm0.35}$ | $\mathbf{69.88}_{\pm1.70}$ |
|  | ✓ | ✗ | $95.99_{\pm2.44}$ | $94.83_{\pm0.52}$ | $68.17_{\pm1.48}$ |
|  | ✓ | ✓ | $97.70_{\pm1.71}$ | $95.68_{\pm0.40}$ | $69.71_{\pm1.92}$ |

Table 28: **Results for `ogbg-molclintox`.**

| Method | Add. Feat. | Virt. Node | ROC-AUC (%) Training | Validation | Test |
|---|---|---|---|---|---|
| GCN | ✗ | ✓ | $83.11_{\pm5.72}$ | $88.78_{\pm1.48}$ | $68.66_{\pm4.95}$ |
|  | ✓ | ✗ | $98.14_{\pm0.91}$ | $99.24_{\pm0.47}$ | $\mathbf{91.30}_{\pm1.73}$ |
|  | ✓ | ✓ | $97.35_{\pm1.30}$ | $\mathbf{99.57}_{\pm0.15}$ | $88.55_{\pm2.09}$ |
| GIN | ✗ | ✓ | $86.12_{\pm5.50}$ | $90.79_{\pm1.10}$ | $61.79_{\pm4.77}$ |
|  | ✓ | ✗ | $96.31_{\pm1.77}$ | $98.54_{\pm0.48}$ | $88.14_{\pm2.51}$ |
|  | ✓ | ✓ | $93.51_{\pm1.78}$ | $99.18_{\pm0.53}$ | $84.06_{\pm3.84}$ |

Table 29: **Results for `ogbg-molsider`.**

| Method | Add. Feat. | Virt. Node | ROC-AUC (%) Training | Validation | Test |
|---|---|---|---|---|---|
| GCN | ✗ | ✓ | $73.82_{\pm1.02}$ | $59.86_{\pm0.81}$ | $\mathbf{61.65}_{\pm1.06}$ |
|  | ✓ | ✗ | $82.74_{\pm2.99}$ | $\mathbf{64.64}_{\pm0.82}$ | $59.60_{\pm1.77}$ |
|  | ✓ | ✓ | $77.50_{\pm2.58}$ | $61.88_{\pm0.89}$ | $59.84_{\pm1.54}$ |
| GIN | ✗ | ✓ | $72.37_{\pm0.78}$ | $59.84_{\pm0.86}$ | $57.75_{\pm1.14}$ |
|  | ✓ | ✗ | $80.13_{\pm2.91}$ | $64.14_{\pm1.24}$ | $57.60_{\pm1.40}$ |
|  | ✓ | ✓ | $76.60_{\pm1.38}$ | $62.41_{\pm0.99}$ | $57.57_{\pm1.56}$ |

Table 30: **Results for `ogbg-molesol`.**

| Method | Add. Feat. | Virt. Node | RMSE Training | Validation | Test |
|---|---|---|---|---|---|
| GCN | ✗ | ✓ | $0.883_{\pm0.096}$ | $1.128_{\pm0.032}$ | $1.143_{\pm0.075}$ |
|  | ✓ | ✗ | $0.629_{\pm0.041}$ | $1.022_{\pm0.034}$ | $1.114_{\pm0.036}$ |
|  | ✓ | ✓ | $0.758_{\pm0.147}$ | $0.991_{\pm0.04}$ | $1.015_{\pm0.096}$ |
| GIN | ✗ | ✓ | $0.746_{\pm0.158}$ | $0.921_{\pm0.045}$ | $1.026_{\pm0.063}$ |
|  | ✓ | ✗ | $0.628_{\pm0.041}$ | $1.007_{\pm0.028}$ | $1.173_{\pm0.057}$ |
|  | ✓ | ✓ | $0.675_{\pm0.131}$ | $\mathbf{0.878}_{\pm0.036}$ | $\mathbf{0.998}_{\pm0.066}$ |

Table 31: **Results for `ogbg-molfreesolv`.**

| Method | Add. Feat. | Virt. Node | RMSE Training | Validation | Test |
|---|---|---|---|---|---|
| GCN | ✗ | ✓ | $1.163_{\pm0.157}$ | $2.744_{\pm0.201}$ | $2.413_{\pm0.195}$ |
|  | ✓ | ✗ | $0.982_{\pm0.109}$ | $2.582_{\pm0.297}$ | $2.640_{\pm0.239}$ |
|  | ✓ | ✓ | $1.219_{\pm0.153}$ | $2.922_{\pm0.185}$ | $2.186_{\pm0.120}$ |
| GIN | ✗ | ✓ | $1.006_{\pm0.225}$ | $2.567_{\pm0.19}$ | $2.307_{\pm0.340}$ |
|  | ✓ | ✗ | $1.205_{\pm0.360}$ | $2.342_{\pm0.378}$ | $2.755_{\pm0.349}$ |
|  | ✓ | ✓ | $0.934_{\pm0.138}$ | $\mathbf{2.181}_{\pm0.205}$ | $\mathbf{2.151}_{\pm0.295}$ |

Table 32: **Results for `ogbg-mollipo`.**

| Method | Add. Feat. | Virt. Node | RMSE Training | Validation | Test |
|---|---|---|---|---|---|
| GCN | ✗ | ✓ | $0.669_{\pm0.058}$ | $0.855_{\pm0.032}$ | $0.823_{\pm0.029}$ |
|  | ✓ | ✗ | $0.662_{\pm0.046}$ | $0.816_{\pm0.024}$ | $0.797_{\pm0.023}$ |
|  | ✓ | ✓ | $0.545_{\pm0.041}$ | $0.766_{\pm0.011}$ | $0.771_{\pm0.016}$ |
| GIN | ✗ | ✓ | $0.488_{\pm0.029}$ | $0.749_{\pm0.018}$ | $0.741_{\pm0.024}$ |
|  | ✓ | ✗ | $0.479_{\pm0.027}$ | $0.742_{\pm0.011}$ | $0.757_{\pm0.018}$ |
|  | ✓ | ✓ | $0.399_{\pm0.023}$ | $\mathbf{0.679}_{\pm0.014}$ | $\mathbf{0.704}_{\pm0.015}$ |

## Footnotes

[7]Recently, some progress has been made to increase the dataset sizes: http://graphlearning.io. Nevertheless, most of them are still small compared to the OGB datasets, and evaluation protocols are not standardized.

[8]Defined by the difference between training and test accuracy.

[9]The GRAPHSAGE architecture is used for neighbor aggregation.

[10]In our preliminary experiments, we used one-hot encodings of species ID as node features, but that did not work well empirically, which can be explained by the fact that the species ID is used for dataset splitting.

[11]Note that the input features here are graph-aware in some sense, because they are obtained by averaging the incoming edge features.

[12]https://arxiv.org/corr/subjectclasses

[13]In practice, the trained models can also be used to predict labels of even non-ARXIV papers.

[14]NODE2VEC is omitted as it is computationally costly on such a gigantic graph.

[15]For each author, we include all the institutions that the author has ever belonged to.

[16]The R-GCN architecture is used for neighbor aggregation.

[17]Here we obtain node embeddings by applying a linear layer to the raw one-hot node features.

[18]Available at `https://archive.org/search.php?query=creator%3A%22Wikidata+editors%22`

[19]Given a fixed 11GB GPU memory budget, we adopt 100-dimension embeddings for DISTMULT and TRANSE. Since ROTATE and COMPLEX require the entity embeddings with the real and imaginary parts, we train these two models with the dimensionality of 50 for each part. On the other hand, on the high-end GPU with 48GB memory, we are able to train all the models with 6× larger embedding dimensionality.

[20]This means that the training MRR is computed by ranking against randomly-selected negative entities without filtering out triplets that appear in KG. The unfiltered metric has the systematic bias of being smaller than the filtered counterpart (computed by ranking against "true" negative entities, *i.e.*, the resulting triplets do not appear in the KG) [14].

[21]Note that our test MRR on `ogbl-wikikg` is computed using only 500 negative entities per triplet, which is much less than the number of negative entities used to compute MRR in the existing KG datasets, such as FB15K and FB15K-237 (around 15,000 negative entities). Nevertheless, ROTATE gives either lower or comparable test MRR on `ogbl-wikikg` compared to FB15K and FB15K-237 [78].

[22]In Table 17, training MRR is lower than validation and test MRR because it is an *unfiltered* metric (computed by ranking against randomly-selected negative entities), and is expected to give systematically lower MRR than the filtered metric (computed by ranking against "true" negative entities, *i.e.*, the resulting triplets do not appear in the KG).

[23]Wu *et al.* [92] originally used a closely-related metric, PRC (Precision Recall Curve)-AUC, but Davis and Goadrich [21] showed that AP is more appropriate to summarize the non-convex nature of PRC.

[24]`https://github.com/github/CodeSearchNet`

[25]The previous works find that the F1 score over sub-tokens is suitable to assess the quality of a method name prediction, as the semantic of a method name depends solely on its sub-tokens. Note that the F1 score does not take the sub-token ordering into account; thus, "`run_model`" and "`model_run`" are considered as exact match.

[26]The inverse edges are also added to allow bidirectional message passing. The edge direction is recorded in the edge features.

[27]Although the F1 score is order-insensitive, in our preliminary experiments, we find that our order-sensitive decoder performs slightly better than order-insensitive decoder (predicting whether each vocabulary is included in the target sequence or not). During training, all the target sequences are truncated to the length of 5 (Covering 99% of the target sequences), and vocabulary size of 5,000 is used for prediction (covering 90% of the sub-tokens in the target sequences). We additionally added one vocabulary "UNK" to handle any rare/unknown sub-tokens. Predicting "UNK" sub-token is counted as false positive when the F1 score is calculated.

[28]The shape of the matrix is the number of data points times the number of tasks. The matrix can be either a PYTORCH tensor or NUMPY array.