[Reviews · NeurIPS 2020]

Review 1

Summary and Contributions: The authors present Open Graph Benchmark called OGB which pave the way for machine learning and data mining researchers. OGB covers a wide range of areas including biological, social, and molecular networks. Different scales of datasets from small to large are included in OGB. OGB is designed to handle different tasks such as node, link, and graph property predictions. Moreover, OGB provide an automatic end-to-end pipeline which facilitates experimental setup and model evaluation. Overall the paper is very well-motivated and easy to follow. The benchmark can be very beneficial for the ML community.

Strengths: The paper proposes a graph benchmark so theoretical aspects are not needed. The strengths of OGB can be categorized as 1- handling different scales of graphs 2- ability to perform various tasks including node, like and graph property prediction. 3- including a wide range of domains.

Weaknesses: Currently, it seems OGB is not able to handle bi-directional graphs.

Correctness: everything sounds correct.

Clarity: The paper is very well-writtten and I believe it will be one of the impactful graph papers in the future.

Relation to Prior Work: Yes the difference with related work is the ability of handle large scale graphs (100+ million nodes and 1+ billion links ) and is clearly mentioned in the paper.

Reproducibility: Yes

Additional Feedback:


Review 2

Summary and Contributions: The paper introduces the Open Graph Benchmark (OGB), a collection of 15 large scale graph datasets covering three different prediction tasks and several application domains. The OGB also specifies dataset splits and evaluation metrics for each of the datasets. In particular OGB contains non-random, domain-specific splits for each dataset that are designed to reflect the non-iid nature of real-world applications. The paper also provides evaluations of some baseline methods on the OGB datasets, in particular showing that there is a large generalization gap between training and test performance if data is not split at random.

Strengths: - The paper provides a collection of datasets for benchmarking GraphML, which could provide a common baseline for future research. - The emphasis of data splitting, an aspect that is neglected in most ML benchmarks, might lead to algorithms that are more relevant for real applications. - The benchmarks are very well specified, including splits and metrics, which is important to ensure comparability across studies. - Curating benchmark datasets is hard and unfortunately undervalued in the ML community.

Weaknesses: - The contribution goes in several directions which makes the paper hard to evaluate; is the main contribution the selection of existing datasets, introducing new datasets or new versions of datasets, the empirical evaluation or the software tooling? - The dataset does not describe existing datasets and benchmarks, and so it is hard to judge the exact differences between the proposed datasets and currently used datasets. A more direct comparison might be useful, and it's not clear why existing, smaller datasets are not included in the collection. - For some of the datasets, it's unclear if or how they have been used or published before. In particular, the datasets from Moleculenet seem to be mostly reproduced, using the splitting strategy that was suggested in their paper, with the modification potentially being addition of new features. - If the selection of the datasets is a main contribution, the selection process should be made more clear. What was the pool of datasets that was drawn from, and how were datasets selected? An example of such a work is the OpenML100 and OpenML CC-18 for classification, see Bischl et. al. "OpenML Benchmarking Suites". or Gijsbers et al "An Open Source AutoML Benchmark" In addition to selection of the datasets, the selection of the splitting procedure and split ratios also seems ad-hoc and is not detailed. - Neither the software package, nor the datasets, nor the code for the experiments has been submitted as supplementary material, and the details in the paper are unlikely to be enough to reproduce the creation of the datasets or the experiments given the datasets. - Given that many methods aim at one of the three tasks, having 5, 6 and 4 datasets for the tasks respectively, might not be enough for a very rigorous evaluation, in particular if some of the datasets are so large that not all algorithms can be used on them. Addendum: Thank you to the authors for their detailed reply. A repository and online platform for reproducing the experiments was provided, and it was clarified that the datasets are substantially novel. Motivations for the number and choice of datasets were given and I updated my assessment to reflect that.

Correctness: I did not find an issue with the methodology; however, I'm not an expert in the methods that were applied. The selection of hyper-parameters for the benchmark seems somewhat ad-hoc and is not clearly motivated (apart from memory requirements). Most of these algorithms have many hyper-parameters and are influenced by random restarts, neither of which is discussed in detail.

Clarity: The paper is well written and easy to follow apart from minor issues mentioned in the comments.

Relation to Prior Work: I think the description of prior work could be much improved. In particular, Moleculenet seems to already provide datasets, metrics, and dataset splits, and this doesn't seem to be clearly discussed. Providing splits, metrics and leaderboards is also clearly reflecting the widely used OpenML platform (Van Schoren "OpenML: Networked science in machine learning.") which should be mentioned as prior work for tabular data (as well as potentially the AutoML benchmark and OpenML CC18 mentioned above). OpenML provides a Python interface orchestrating experimentation similar to the one described in this work. While the curation of a set of datasets is clearly an added value to the community, the exact contributions should be made more obvious.

Reproducibility: No

Additional Feedback: I find sales rank an odd splitting scheme for products, as well-performing products often behave very differently and are usually much fewer. I would provide the domain of each dataset in one of the overview tables. It's not mentioned in the main text that the OGB Pipeline takes the form of a Python library. This should be made more clear early on. If the python library is to be considered a major contribution, I would have expected at least some explanation of a standard usecase in the paper. Most graphs have a MaxSCC ratio close to 1, meaning they are (nearly) strongly connected. Wouldn't it be natural to exclude the remaining nodes as outliers? It's not clear whether previous datasets such as Cora and Citeseer are also available via the Python interface. The use of Precision-Recall Curve (PRC)-AUC is unclear to me. Either a lower step approximation can be used to compute the area, in which case this would be Average Precision, or a linear interpolation could be used, which is not appropriate for PR curves (see Davis and Goadrich: The Relationship Between Precision-Recall and ROC Curves). So the meaning of the term should be clarified, and linear interpolation should not be used with PR curves. The claim "Overall, OGB provides the meaningful data splits and improved molecular features, together with their data-loaders, making the evaluation and comparison on MOLECULENET datasets easier and standardized." seems shaky to me. From what I understand, moleculenet provides all of these apart from the improved features. The deepchem package (https://github.com/deepchem/deepchem) should maybe be referenced in this context. Also, there is a spurious "the" before meaningful. The citations for the supplementary material seem to be missing, which makes it hard to follow the appendix.


Review 3

Summary and Contributions: This paper presents a new set of realistic benchmarks for graph machine learning research, called open graph benchmark (OGB). Some graphs in OMG are significantly larger than existing datasets, and cover a variety of domains, including social networks, biological networks, molecular graphs, and knowledge graphs. In addition, the authors propose different splitting between training, validation, and testing for some existing datasets. These splittings are meant to be more realistic and increase the challenge of the tasks which are evaluated on these datasets One of the OMG goals is to facilitate reproducible graph ML research. To this end, the papers provides an extensive empirical analysis for several datasets, and describes an e2e ML pipeline that standardizes (a) data loading, (b) experimental setup, and (c) model evaluation. The datasets and the OMG pipeline are publicly available.

Strengths: - Provide a set of benchmarks for graph ML research that go beyond the existing benchmarks both in terms of scale and diversity. - Provide a graph ML analytics pipeline that enables researchers to easily test new algorithms and reproduce previous results. - Provide preliminary evaluations on the proposed datasets of a variety of graph ML algorithms,

Weaknesses: The authors oversell a bit their contributions. - Quite a few datasets have been published before (e.g., ogbn-products, gbn-proteins, ogbl-ppa), and in those cases the main contribution is better, "more realistic" splits. I'd make this more clear in the introduction. - One of the big claims of the paper is that it contains graphs that are orders of magnitude, more precisely 100M+ vertices and 1B+ edges. While this is true, only one graph (i.e., papers100M) has this property. Next in size is ogbn-products which was published before. I'd be more precise when quoting the size of the graphs in OGB compared to the graphs in previous datasets. In summary, I do believe that the proposed benchmarks and the graph ML pipeline described in this paper have the potential to advance the research in graph ML processing, maybe the same way as other benchmarks (e.g., CIFAR ImageNet) advanced the research in other areas of ML.

Correctness: Yes, I believe all claims in the paper are correct.

Clarity: Yes, the paper is clearly written.

Relation to Prior Work: Yes, it is, though some claims about the novelty of the datasets could be tone down a bit (see above).

Reproducibility: Yes

Additional Feedback: This is a well written paper, and the appendix is comprehensive. In addition, to the weaknesses mentioned above, I have only a few minor comments. First, in many cases your main contribution when it comes to datasets is using different splits for already available datasets. Please comment why the particular splits For instance why use the 10% of the top selling products for training, and the next 2% for validation in the case of the ogbn-products graph? Did you try other splits? Also, what are the actual splits for gbl-wikikg and ogbg-molhiv? Second, please provide more details about the OBG pipeline. In particular, can you provide some performance numbers for data loading and some training times? Also, of curiosity, how many lines of code?

[Author Response · NeurIPS 2020]

We thank the reviewers for their time and valuable feedback. Overall, we are glad that the reviewers found OGB to be
important to advance the field of graph ML. Below, we clarify a number of important points raised by the reviewers: **(1)**
**originality of datasets**, **(2) datasets selection criteria**, **(3) core contribution**, **(4) reproducibility**, and **(5) others**.

**(1) Originality of datasets.** R4 and R5 raised a critical concern that many of our datasets are simply reproduction
of existing ones. This is not true for two reasons: First, nearly all (except `ogbn-products` and `ogbg-mol*`)
datasets are in fact constructed by ourselves from public raw data, together with domain experts. Thus, the majority
of our OGB datasets (12 out of 15) are completely new and original (note that `ogbg-ppa` and `ogbn-proteins`
are also original and different from the existing PPI graph benchmark. Details in L643–L653 in Appendix). Sec-
ond, although the graphs of `ogbn-products` and `ogbg-mol*` are indeed defined by existing works (which
we will emphasize more in our final version), we identified and resolved some important problems around data
splitting. Specifically, for `ogbn-products`, existing work [15] did not define a validation set and used a ran-
dom split with the large training proportion (90%), yielding almost no generalization gap (L167–L169). Even
for MOLECULENET, we noticed a serious problem that the scaffold split is not standardized (*e.g.*, "scaffold split"
used in [92] is different from [36,40] as well as recent arXiv:2007.02835), because how different scaffolds are
put into different splits is quite arbitrary (L689–L693 in Appendix). We think resolving these issues and stan-
dardizing the evaluation procedure is important. We will clearly mention this contribution in our final version.

**(2) Datasets selection criteria.** R4 questioned whether we have clear criteria
to select the 15 datasets, similarly to those in OpenML-CC18. Indeed, we
do have such criteria: we ensure the diversity of task categories, scales (as
defined in L63–L69), and domains, as illustrated in Figure 1 of the paper.
Consequently, our 15 datasets are indeed diverse, as shown in the green cells in
the right table, resulting in diversity in graph structure (Section 2). OGB is an
on-going community-driven effort, and we are constructing additional datasets
to fill in the grey blanks so that OGB datasets are maximally diverse.

| Task | Node property prediction ogbn- | | |
|---|---|---|---|
| **Domain** | Nature | Society | Information |
| Small | | arxiv | |
| Medium | proteins | products | mag |
| Large | | papers100M | |

| Task | Link property prediction ogbl- | | |
|---|---|---|---|
| **Domain** | Nature | Society | Information |
| Small | ddi | collab | biokg |
| Medium | ppa | citation | wikikg |
| Large | | | |

| Task | Graph property prediction ogbg- | | |
|---|---|---|---|
| **Domain** | Nature | Society | Information |
| Small | molhiv | | |
| Medium | molpcba / ppa | | code |
| Large | | | |

We would also like to clarify fundamental differences between OGB and
OpenML-CC18. First, OGB actually *constructs* most of the datasets from
scratch, while the OpenML selected existing datasets. Second, OGB exten-
sively evaluates and tests the datasets to make sure they are appropriate, well-behaved and robust. Third, OGB
deliberately provides a small selected set of challenging large-scale datasets (around 5 for each task category), as
opposed to OpenML-CC18 that provides many small-scale datasets (72 datasets with 500 to 100K samples). The benefit
of the former benchmark (OGB) is to allow the community to really focus on challenging problems and easily compare
different models on a few benchmarks (similar to ImageNet, CIFAR10, and SQuAD), which is why we did not include
many existing small-scale datasets, such as Cora and CiteSeer (extensive discussion on existing datasets is provided in
Appendix A, due to space constraint). Our design principle is in contrast to the OpenML-CC18 and the TU datasets [43]
(a collection of more than 50 small-scale non-original graph classification datasets); which inevitably causes different
works to evaluate models on different subsets of the datasets, making it hard to compare performance across papers.

**(3) Core contribution** As R4 nicely pointed out, OGB indeed has contributions to many directions, but as our paper
title suggests, our main focus is on introducing and defining a set of realistic, large-scale, and challenging datasets and
tasks, many of which are our original ones. We also perform extensive baseline experiments and provide easy-to-use
code (as done by *e.g.*, MS-COCO and the OpenML), with the goal of analyzing how existing models perform on our
newly-introduced tasks and making our new datasets easily accessible to users (in the same spirit as the OpenML).

As suggested by R4, in the final version, we will cite the OpenML and OpenML-CC18 and carefully discuss the above
(1)–(3), clarifying our exact contribution.

**(4) Reproducibility and code.** R4 raised an important concern about reproducibility, which we agree is crucial for
OGB. To address this, we have provided to our Area Chair the link to our anonymized Github repository, which contains
all of our package scripts and baseline code (note that external links are not allowed to be included in our response here).
Regarding the package description, it is provided with example code snippets in Appendices E.1 and E.2, due to space
constraints. The data loading and training performance are the same as PyG and DGL, which is highly efficient but the
exact number depends on the dataset sizes (*e.g.*, loading a processed medium-scale dataset takes about 5 seconds).

**(5.1) Sales ranking split of `ogbn-products`.** R4 and R5 raise concerns
about the split used in `ogbn-products`. We did try different split ratios, and
selected the current one to ensure the training nodes are not too skewed (as
pointed out by R4) in the sense that the class balance is almost the same across
training/validation/test splits. Also, we think 10% for training is an appropriate
number to simulate a realistic transductive semi-supervised learning setting.

| Method | Additional Features | Virtual Node | AP (%) | | |
|---|---|---|---|---|---|
| | | | Training | Validation | Test |
| GCN | ✗ | ✔ | 36.25±0.71 | 23.88±0.22 | 22.91±0.37 |
| | ✔ | ✗ | 28.04±0.58 | 20.59±0.33 | 20.20±0.24 |
| | ✔ | ✔ | 38.25±0.50 | 24.95±0.42 | 24.24±0.34 |
| GIN | ✗ | ✔ | 45.70±0.61 | 27.54±0.25 | 26.61±0.17 |
| | ✔ | ✗ | 37.05±0.31 | 23.05±0.27 | 22.66±0.28 |
| | ✔ | ✔ | 46.96±0.57 | **27.98**±0.25 | **27.03**±0.23 |

**(5.2) PRC-AUC of `ogbg-pcba`.** We thank R4 for pointing out the issue with PRC-AUC. We now use the suggested
Average Precision (AP), and observed the same trend (see Table above). We will update this in the final version.

[Meta-Review · NeurIPS 2020]

The paper introduces open graph benchmark, a collection of 15 datasets, as well as software for reproducible benchmarking. Three knowledgeable reviewers support acceptance for the contributions of the paper, and I also recommend to accept for NeurIPS 2020. However, I encourage the authors to address the concerns raised by the reviewers in the final version, in particular to reword the claims of the paper as mentioned by Reviewers #4 and #5.